# Was there a volcanic induced long lasting cooling over the Northern Hemisphere in the mid 6th-7th century?

Evelien van Dijk[1], Johann Jungclaus[2], Stephan Lorenz[2], Claudia Timmreck[2], and Kirstin Krüger[1]

[1]Department of Geosciences, University of Oslo, Oslo, Norway
[2]Max Planck Institute for Meteorology, Hamburg, Germany

**Correspondence:** Evelien van Dijk (evelien.van.dijk@geo.uio.no) and Kirstin Krüger (kirstin.krueger@geo.uio.no)

**Abstract.** The climate in the Northern Hemisphere (NH) of the mid-6th century was one of the coldest during the last two millennia based on multiple paleo-proxies. While the onset of this cold period can be clearly connected to the volcanic eruptions in 536 and 540 Common Era (CE), the duration, extent and magnitude of the cold period is uncertain. Proxy data are sparse for the first millennium, which compounds the uncertainties of the reconstructions. To better understand the mechanisms of the
prolonged cooling, we analyze new transient simulations over the CE and enhance the representation of mid 6th to 7th century climate by additional ensemble simulations covering 520-680 CE. We use the Max Planck Institute Earth System Model. To apply the external forcing as recommended in the Paleo Model Intercomparison Project, Phase 4.

After the four large eruptions in 536, 540, 574 and 626 CE, a significant mean surface climate response in the NH lasting up to 20 years is simulated. The 2 m air temperature shows a cooling over the Arctic in winter, corresponding to the increase
in Arctic sea-ice, mainly in the Labrador sea and to the east of Greenland. The increase in sea-ice extent relates to a decrease in the northward ocean heat transport into the Arctic within the first two years after the eruptions, and to an increase in the Atlantic Meridional Overturning Circulation, which peaks 10 years after the eruptions. A decrease in the global ocean heat content is simulated after the eruptions that does not recover during the simulation period. These ocean – sea-ice interactions sustain the surface cooling, as the cooling lasts longer than is expected solely from the direct effects of the volcanic forcing,
and are thus responsible for the multidecadal surface cooling. In boreal summer, the main cooling occurs over the continents at mid-latitudes. A dipole pattern develops with high sea level pressure and a decrease in both precipitation and evaporation poleward of 40°N. In addition, more pronounced cooling over land compared to ocean leads to an enhanced land-sea contrast. While our model ensemble simulations show a similar $\sim 20$ year summer cooling over NH land after the eruptions as tree ring reconstructions, a volcanic induced century long cooling, as reconstructed from tree ring data, does not occur in our
simulations.

## 1   Introduction

Large volcanic eruptions are the major driver of natural climate variability in the pre-industrial era of the last millennium (Hegerl et al., 2006; Schurer et al., 2014). To assess what potential impact they could have on future climate, it is important to understand the climate response to volcanic forcing in the past, and which mechanisms were involved. In addition, volcanic

eruptions are often studied as a natural analog for solar radiation management (Robock et al., 2013).

Several cluster eruptions and volcanic double events occurred in the last 2000 years as recorded in the ice core record, which coincide with cold periods in Northern Hemisphere (NH) tree ring records (Briffa et al., 1998; Sigl et al., 2013). One of the coldest decades of the last 2000 years in the NH and Europe is visible in tree rings during the mid-sixth century, which was
preceded by two volcanic eruptions as recorded in ice cores (Larsen et al., 2008; Büntgen et al., 2011). Furthermore, historic documents reported a dimming of the sun in 536 CE (Stothers, 1984; Rampino et al., 1988). Revised ice core chronologies reveal two sulfate peaks that correspond to eruptions in 536 CE and 540 CE followed by two large eruptions in 574 CE and 626 CE (Baillie, 2008; Sigl et al., 2015). Reconstructed tree ring temperatures from the Alps and Altai show a century-long cooling that might have exceeded that of the Little Ice Age (LIA) during the 14th-19th century (Büntgen et al., 2016). Based on these
records, this period was called the Late Antiquity Little Ice Age (LALIA). However, other studies reveal contrasting results on how long lasting the surface cooling in the NH extratropics was, varying from multi-decadal to centennial cooling. These results are based on tree ring reconstruction methods and tree ring record updates, as well as ice-core records and documentary evidence (e.g., Esper et al., 2012b; Matskovsky and Helama, 2014; Helama et al., 2017; Guillet et al., 2020; Büntgen et al., 2021; Helama et al., 2021). Thus, the duration of this volcanic induced cooling event remains open.

Climate models can not only shed light on the climate response to volcanic eruptions, but also on the underlying mechanisms and feedbacks. If consecutive eruptions occur within a few years, ocean - sea-ice feedbacks may maintain the surface cooling over longer periods, up to centennial time scales (McGregor et al., 2015; Lee et al., 2021). With the help of climate models we can test if a series of volcanic eruptions may have caused severe long-lasting cooling over the NH and Europe during the
mid-6th to 7th century. Previous model studies have simulated the surface climate response to volcanic eruptions in the last millennium and found up to a decade of surface cooling due to extremely large eruptions (Stenchikov et al., 2009; Timmreck et al., 2009; Guillet et al., 2020).

A series of decadal-paced volcanic eruptions in the mid-13th century has been suggested to have caused the onset of the LIA (Schneider et al., 2009; Zhong et al., 2011; Miller et al., 2012) and ocean - sea-ice feedbacks have been connected to this cool-
ing (Lehner et al., 2013; Moreno-Chamarro et al., 2017). However, hardly any modeling studies investigating volcanic-climate impacts during the first millennium exist. Toohey et al. (2016a) carried out climate simulations from 536-550 CE, analyzing the effect of the 536/540 CE volcanic double event on the NH surface climate. The volcanic forcing was reconstructed based on the sulfate deposition in ice cores and the aerosol climate model MAECHAM-HAM (see Table 1), which was used as input for the Max Planck Institute Earth System Model (MPI-ESM) climate simulations. They found decreased NH temperature
anomalies of up to 2 K and increased Arctic sea-ice after the volcanic double event lasting the entire simulation period of 15 years. However, this simulation period was too short to study multidecadal cooling. In addition, the simulations from Toohey et al. (2016a) were initialized using a pre-industrial conditions control run, whereas our simulations include the forcing history of the first half-millennium.

The aim of this study is to investigate whether a series of volcanic eruptions induced a multidecadal to centennial cooling in the mid-6th to 7th century. We performed 160 yearlong (520-680 CE) MPI-ESM ensemble simulations branched off of one Paleoclimate Modeling Intercomparison Project phase 4 (PMIP4) past2k simulation (Jungclaus et al., 2017). We focus on the NH extratropical climate in comparison with available temperature proxy reconstructions. Multiproxy reconstructions in the NH during this period consist of mainly tree ring, marine sediments, and ice core records. Marine sediments have a

coarse chronological resolution and would therefore not capture the volcanic signal, and ice core records are confined to the Greenland Icesheet (PAGES Consortium et al., 2017). Tree ring records have an annual resolution with an absolute dating. Therefore, we use the most recent tree ring ensemble reconstruction for the past2k CE (Büntgen et al., 2021) next to other reconstructions (i.e., Stoffel et al., 2015; Guillet et al., 2020) to compare to the model simulations in this study. In contrast to the study of Toohey et al. (2016a), the short term (annual), as well as the long term (decadal to centennial) effects of the

536/540 CE volcanic double event plus the other eruptions during 520-680 CE on the coupled climate system are analyzed. In addition to the surface climate and sea-ice impacts, we also study changes in atmospheric and ocean circulation, hydrology and ocean - sea-ice feedbacks.

In Section 2, the model details, set up and experiments, as well as tree ring data are described. In Section 3, the model results are presented and discussed. Finally, a summary and conclusion are given in Section 4.

## 2  Methods

**Model and experiment description**

**The MPI-ESM model**

The MPI-ESM1.2.01p5 used in this study is the low-resolution (LR) version used for the Coupled Model Intercomparison Project phase 6 (CMIP6) and the PMIP4. The atmosphere component (ECHAM6.3, Stevens et al., 2013) has a horizontal

resolution of 1.9°x 1.9°and 47 vertical layers (T63L47) up to 0.01 hPa or 80 km altitude. The ocean component (MPIOM 1.6 GR1.5/L40, Jungclaus et al., 2013) features a conformal mapping grid with a nominal resolution of 1.5°. The grid poles are placed over Greenland and Antarctica and the actual horizontal resolution ranges from 22 km near Greenland to roughly 200 km in the Pacific Ocean. The vertical grid is represented on 40 unevenly spaced z-levels, with 20 levels in the upper 700 m. A complete description of MPI-ESM1.2 in its CMIP6 configurations, including parameter and tuning choices, is given by

Mauritsen et al. (2019).

**The PMIP4 - past2k simulations**

For its fourth phase, PMIP has selected four experiments as contribution to CMIP6 (Kageyama et al., 2017). In addition to time-slice experiments, the transient simulation over the last millennium "past1000" was chosen as a core experiment. The "past1000" experimental protocol (Jungclaus et al., 2017) describes several extensions to the standard last millennium

simulation and includes the tier-3-category "past2k" experiment that covers the entire CE. The aim of the "past2k" experiment

is to extend the scope of the PMIP transient simulations further back into the past and to encourage model-data comparison studies taking full advantage of the past global changes (PAGES) "PAST2K" data base (Martrat et al., 2019). The MPI-ESM experiments described here are, to our knowledge, the first past2k runs that are fully compliant with the CMIP6 and PMIP4 protocols. The two simulations started from different dates taken from the end of a 1200-year long "spin-down" simulation under constant 1 CE boundary conditions, which was initialized from the MPI-ESM-LR piControl simulation for CMIP6. Only one of these simulations included all the output options necessary to produce CMOR-compliant output (Eyring et al., 2016) for publication on the Earth System Grid Federation (ESGF). This past2k run is referred to as past2k "run 1" from here on, and the other past2k run is referred to as past2k "run 0".

**The 520-680 CE simulations**

For this study, we ran ten ensemble members, covering the period 520-680 CE. The simulations are branched off one of the two past2k runs (run 0) in the year 521 by perturbing the climate. For each ensemble member the atmospheric diffusivity was changed by $1 \cdot 10^{-5} \ m^2 s^{-1}$ to simulate slightly different climate states by the year 536 CE, the year of the first large volcanic eruption. In earlier studies, the initial state is taken from a pre-industrial control simulation. In contrast our approach allows us to include the climate and forcing history of the previous decades and centuries as well as their integrative effects (Gleckler et al., 2006). The simulations were run with the new volcanic forcing for PMIP4, as well as best estimate conditions for the 6th-7th century (see external forcing section). For the ensemble mean results (n=12), we included ten ensemble members from the 520-680 CE and two ensemble members from the PMIP4 past2k simulations. The anomalies were calculated by subtracting the mean of 1-1850 CE from the past2k "run 0" from the mean of the volcanic ensemble. We use the mean over 1-1850 CE to have a reference climate that is representative for the entire pre-industrial CE. We also tested different anomaly calculations with a reference period ten years before the 536 CE eruption, as well as 1200 years of the control run, both without volcanic eruptions, and the resulting temperature anomalies are within +/- 0.15 K. The significance level was calculated from the 1200-year control run. To account for the different response times of two and 20 years, the two and 20 year means were taken from the 1200 year control run using bootstrapping. As the mean response is for four eruptions, four random time steps were taken from the mean control time series to account for this. To increase the power of the test, this was iterated 1000 times for each variable, and from these 1000 samples a new time series was created. The standard deviation was then calculated from this random time series. The one or two sigma were calculated for the significance levels.

**External forcing**

The external forcing for the MPI-ESM simulations follows the protocol by Jungclaus et al. (2017) using the respective default choices for the past1000 experiment. Except for land-use/land-cover change data, all external forcing agents described in Jungclaus et al. (2017), see below, are available for the entire CE. Total solar irradiance and spectral solar irradiance are derived from cosmogenic radio isotopes through a chain of physics-based models (Vieira et al., 2011; Usoskin et al., 2014). We chose here the 14C-based SATIRE-M version (see Section 3). The effect of solar-derived ozone changes is parameterized by a simple scaling scheme. Greenhouse gases ($CO_2, CH_4, N_2O$) follow the recommendations for CMIP6 (Meinshausen et al., 2016),

and the orbital parameters are calculated internally in ECHAM6's calendar routine. The CMIP6 land-use and anthropogenic land-cover changes have been reconstructed for the period 850 to 2014 CE (Jungclaus et al., 2017; Hurtt et al., 2020). The data set is based on modern estimates of division into several agricultural uses and urban areas coming from the Historical Land Use Data Set for the Holocene (HYDE3.2, Klein Goldewijk, 2016). Considering several options (e.g. linear ramp-up) we prescribed a constant land-cover for the first 850 years of the past2k runs.

**Volcanic forcing**

The volcanic forcing used for the runs is described by Jungclaus et al. (2017) with details on the eVolv2k data set being described by Toohey and Sigl (2017). eVolv2k is used in combination with the Easy Volcanic Aerosol module (EVA), as described by Toohey et al. (2016b). The eVolv2k includes estimates of stratospheric sulfur injections (VSSI) and the locations of the eruptions (latitudes) based on ice core data from Antarctica and Greenland. A calibration factor is used to convert the ice core deposition value to stratospheric sulfur loading (Gao et al., 2007). The details of the eruptions occurring in the period of the ensemble runs are given in Fig. 1. The eruptions are set to January in the model forcing as the actual eruption month is unknown. Each eruption gives an abrupt increase in aerosol optical depth (AOD), and the differences between them are distinct. In this study, we focus on the four largest eruptions out of the 14 in the study period, where 536 CE and 626 CE are NH extra-tropical eruptions and 540 CE and 574 CE eruptions are tropical eruptions (Table 1). For the NH extra-tropical eruptions, the aerosols are mainly contained within the NH hemisphere, whereas for the tropical eruptions the AOD spreads out over both hemispheres. Figure 1b shows the accumulated AOD per latitude for 15 year intervals, where 535-550 CE stands out as the period with the strongest forcing of the 160-year simulation period.

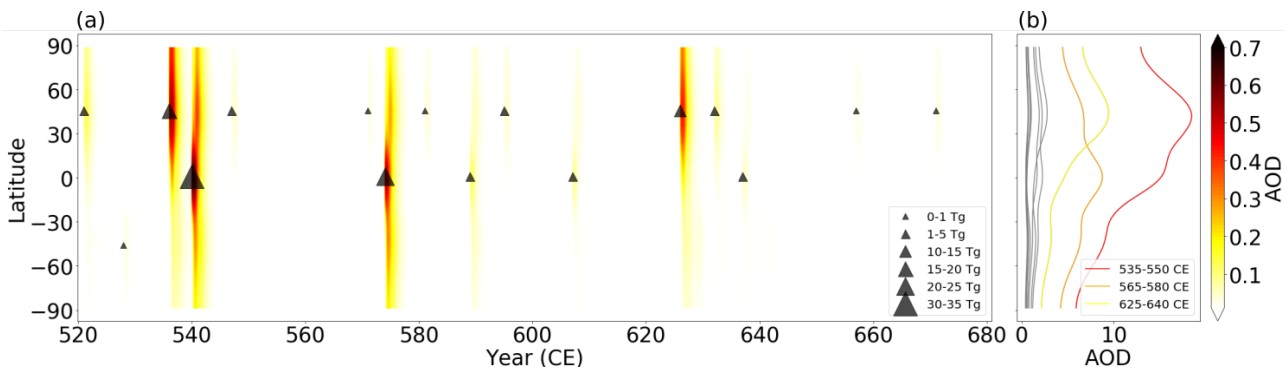

**Figure 1.** a) Zonal mean volcanic forcing (Aerosol Optical Depth, AOD) for the study period (520-680 CE) and b) zonal mean accumulated AOD (520-680 CE) in 15 year bins from the reconstructed volcanic forcing of Toohey and Sigl (2017). The three 15 year periods with the highest volcanic forcing are highlighted in color and labeled. The triangles in (a) highlight the zonal locations and different strength in terms of stratospheric sulfur injection of all volcanic eruptions as specified in the legend (Tg = $10^{12}$ g).

**Table 1.** Volcanic forcing and stratospheric sulfur injection for eruptions used in this study (based on eVolv2k; Toohey and Sigl, 2017). Details for the Toohey et al. (2016a) 536 and 540 CE reconstruction based on MAECHAM-HAM are added. The four, respectively, two largest eruptions are highlighted as bold text.

| Eruption year | Eruption month | Eruption latitude | S [Tg] | Max global AOD | Reference |
|:---:|:---:|:---:|:---:|:---:|:---:|
| 521 | Jan | NH extratropical | 3.7 Tg | 0.07 | Toohey and Sigl (2017) |
| 528 | Jan | SH extratropical | 0.8 Tg | 0.01 | |
| **536** | **Jan** | **NH extratropics** | **18.8** | **0.29** | |
| **540** | **Jan** | **Tropics** | **31.8** | **0.38** | |
| 547 | Jan | NH extratropical | 1.1 Tg | 0.02 | |
| 571 | Jan | NH extratropical | 0.7 Tg | 0.01 | |
| **574** | **Jan** | **Tropics** | **24.1** | **0.31** | |
| 581 | Jan | NH extratropical | 0.9 Tg | 0.02 | |
| 589 | Jan | Tropical | 4.4 Tg | 0.07 | |
| 595 | Jan | NH extratropical | 1.1 Tg | 0.02 | |
| 607 | Jan | Tropical extratropical | 2.7 Tg | 0.04 | |
| **626** | **Jan** | **NH extratropics** | **13.2** | **0.21** | |
| 632 | Jan | NH extratropical | 2.1 Tg | 0.04 | |
| 637 | Jan | Tropical | 1.7 Tg | 0.03 | |
| 657 | Jan | NH extratropical | 0.7 Tg | 0.01 | |
| 671 | Jan | NH extratropical | 0.8 Tg | 0.01 | |
| **536** | **March** | **42°N** | **15** | **0.9** | Toohey et al. (2016a) |
| **540** | **Jan** | **15°N** | **25** | **0.9** | |

**Tree ring data**

For the model-tree ring temperature comparison, different tree ring data and reconstructions are used. The tree ring data used are based on tree ring width (TRW) and maximum latewood density (MXD). TRW is known to have biological memory and gives a lagged and smoothed response to volcanic eruptions. In contrast, MXD is based on cell density, which gives a better

representation of volcanic induced surface cooling (Anchukaitis et al., 2012; Esper et al., 2015; Zhu et al., 2020; Ludescher et al., 2020). MXD data is therefore the preferred target for our model comparison if available. However the data is sparse during the first millennium. Therefore, both tree ring methods are taken into account using the reconstructions from Stoffel et al. (2015), Guillet et al. (2020), and Büntgen et al. (2021), next to others (see Appendix A). Stoffel et al. (2015) and Guillet et al. (2020) consist of a mix of MXD and TRW. The tree ring ensemble reconstruction from Büntgen et al. (2021) is based on TRW only and is taken from nine sites over the NH covering the past 2000 years (Fig. A1). The raw data were distributed to 15 different dendrochronology groups. These groups all carried out their own statistical methods on the data, after which the now different data sets were combined to form a tree ring ensemble. More details about the data and the reconstruction method can be found in the corresponding publications.

To use the same reference period for the model and tree ring data, we subtracted the 1-1850 CE mean from the model and tree ring ensemble. For the model-tree ring comparison a land mask was applied to the model 2 m air temperature and we analyzed only the NH extratropics between 40° and 75°N. The tree ring data sets capture the boreal summer temperature during June, July and August (JJA) and were therefore compared to JJA 2 m air temperatures from the model.

## 3  Results

**Volcanic response**

Figure 2 displays the evolution of NH 2 m air temperature for the ensemble in comparison with the entire past2k simulations. The variations in the ensemble simulations fall within the variability of the past2k runs and the volcanic signals are significant and clearly stand out (Fig. 2a). The four large eruptions in the period 520-680 also clearly stand out in the yearly mean NH 2 m air temperature, precipitation and sea-ice area anomalies and the ocean heat content for the upper 700 m (Fig. 2). Variations in the solar forcing are very small compared to the volcanic forcing and cannot explain these distinct deviations. Following the 540 CE eruption, maximum deviations are reached during the 520-680 period, with a peak cooling of the NH 2 m air temperature of $\sim$2 K, precipitation decrease of 0.2 mm/day, ocean heat content decrease in the upper 700 m of $1.5 \cdot 10^{23}$ Jm$^{-1}$. Additionally an increase in the Arctic sea-ice area of $1.5 \cdot 10^{12}$ m$^2$ compared to the 1-1850 CE mean.

The responses in surface climate show a longer recovery than the AOD, indicating that additional processes play a role in cooling the surface climate after volcanic eruptions. The AOD peaks after $\sim$12 months and is back at 0 after 3-4 years (Fig. 2b). The temperature (Fig. 2c) reveals a maximum cooling in the first and second year after the eruption and is decreased for 20 years after the 540 CE, for 30 years after the 574 CE and for 14 years after 626 CE eruptions, much longer than the direct response of the AOD. The temperature decrease and sea-ice extent increase (Fig. 2e) peak simultaneously, and show a similar long recovery after the eruptions. The precipitation decrease follows a similar pattern as temperature and sea-ice area changes, but with a shorter recovery period. The precipitation anomalies are back to zero after 10-15 years after the eruptions. The global ocean heat content anomaly for the upper 700 m (Fig. 2f) shows a maximum decrease immediately after the eruptions, which is significant for 30-40 years, and does not return to 0 anomaly during the entire study period. The global ocean heat content is the only variable that reveals a century long anomaly due to the subsequent eruptions. The Atlantic ocean heat trans-

port at 60°N reflects high inter-annual variability, which makes it hard to distinguish any significant volcanic signal. After all

180 large volcanic eruption we diagnose a robust decrease in northward heat transport, even though it rarely exceeds the 2-sigma threshold derived from the control simulation. (Fig. 2g). Towards the end of the simulation period the ensemble shows a larger spread in the ocean heat content than at the beginning of the simulations.

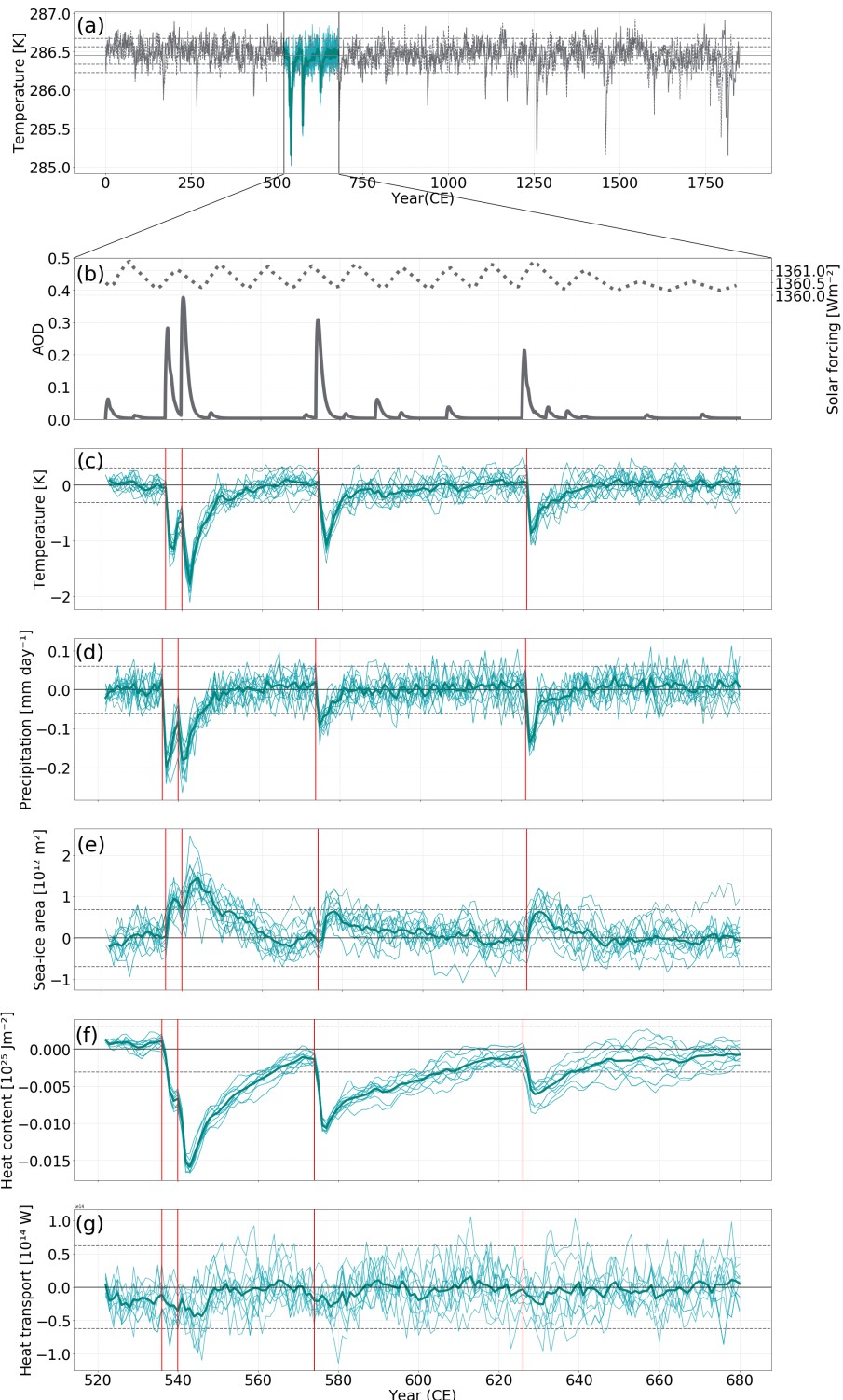

**Figure 2.** Time series of a) NH 2 m air temperature for the two past2k experiments, b) global AOD volcanic forcing and solar forcing. Time series for the MPI-ESM 521-680 CE ensemble runs of c) NH 2 m air temperature, d) NH total precipitation, e) Arctic sea-ice area, f) global ocean heat content for the upper 700 m, and g) Atlantic ocean heat transport at 60°N. All variables are given as anomalies wrt 1-1850 CE. The timing of the volcanic eruptions in the study period (520-680 CE) is represented by a red line. Ensemble runs are blue and the ensemble mean is the thick blue line. The grey dashed lines indicate the $2\sigma$ of the control run.

## Surface climate response

To get more insights into the surface climate response we investigate the spatial patterns of the mean short-term (2 years) and long-term (20 years) response following the four large eruptions in Fig 3. The maps show anomalies wrt 1-1850 CE for boreal winter (DJF) and summer (JJA), where the first two winters after the eruption correspond to year 1 and year 2.

For boreal winter, the short term response shows the most pronounced near-surface cooling over the Hudson Bay, Labrador Sea, Baffin Bay, and surrounding Western, Southern and Eastern Greenland. The sea level pressure (SLP) anomalies reveal increased pressure over high latitudes poleward of 40-50°N, except for Greenland. At the same time decreased pressure is visible over the Northern Pacific indicating an eastward shift of the Aleutian low and corresponding wind changes. The 10 m wind anomaly shows increased westerly flow over the North Atlantic north of 50°N, and a reduction in westerly winds south of ~40 °N. The latitudes north of 40°N show a decrease in precipitation, except for a small area off the west-coast of Norway, whereas south of this latitude precipitation increases. For evaporation there is a land-sea contrast, where south of 40°N there is a decrease over the oceans and an increase over land. The opposite pattern emerges north of 40°N, with the largest increase in evaporation occurring off the east-coast of Japan. The cloud cover fraction anomaly patterns follow those of precipitation and evaporation closely (not shown).

The boreal summer pattern for two years after the eruptions is very similar for 2 m air temperature and precipitation. The 2 m air temperature is decreased everywhere over the NH, but the cooling is strongest south of ~45°N over the continents (Fig. 3a), which corresponds to the maximum in AOD (Fig. 1). This separation between south and north of 45°N is also visible in the hydrology, where south of 45°N precipitation and evaporation over land increase while they decrease north of it. In summer, the sign of the evaporation anomalies over land and ocean is opposite. The SLP in summer is increased over the polar region and over the continents, except for some areas on the western side in North America and Southern Europe. Over the North Atlantic and the North Pacific, there is a decrease in SLP south of 45°N, with the opposite signal in the SLP anomalies north of this latitude, reflecting an atmospheric circulation separation. The wind anomalies follow the SLP anomaly patterns, with anomalous westerly flow around the decreased low pressure over the Arctic and an anomalous cyclonic pattern around the decreased high pressure systems over the oceans. Wind anomalies above 0.5 m/s only appear over the oceans and coastal regions.

In summary, the SLP, precipitation and evaporation maps show a dipole pattern over the NH with a separation at ~40°N in winter and ~45°N in summer, indicating atmospheric circulation changes after two years of the large volcanic eruptions. The wind anomalies show increased surface westerlies north of 60°N, and decreased westerlies south of 50°N over the North Atlantic in NH winter. To get a better insight in the response after each large eruption, the NAO index (see Fig. A4 and corresponding text) is given for the entire study period, as well as an epoch analysis for up to 20 years after the four large eruptions. All except the 536 CE eruption show a positive NAO the first winter after the eruption (year 1) followed by a weak positive or a negative second winter. This leads to a weak positive NAO response in the 2-year ensemble mean (Fig. 3).

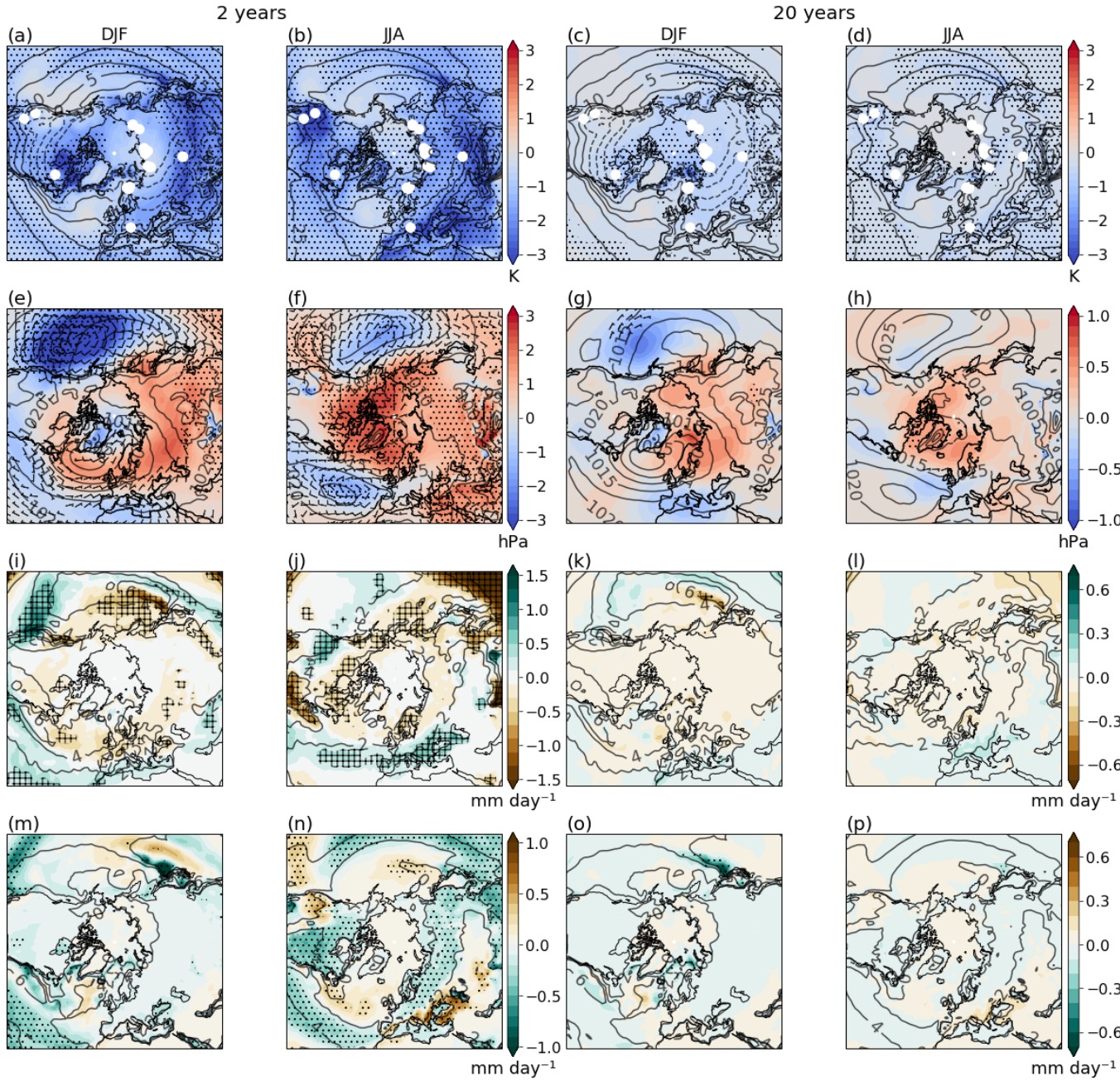

**Figure 3.** NH maps of boreal winter (DJF) and summer (JJA) 2 m air temperature (a-d), sea level pressure and 10 m wind (e-h), precipitation (i-l), and evaporation (m-p) for 2 years and 20 years after the eruptions, poleward of 30°N. The maps represent the ensemble mean of the 2 and 20 year mean after the four major eruptions in the study period. The two years after the eruption are year 1 and year 2 after the eruption for DJF, and the year of the eruption and one year after for JJA, respectively. All variables are given as anomalies wrt 1-1850 CE. The 1-1850 CE climatology is given as contours and the tree ring locations are represented by white dots in the 2 m air temperature maps. The ensemble standard deviation $1\sigma$ ($2\sigma$) for 2 m air temperature, SLP, evaporation and precipitation are hatched (stippled). Note the different scales for the 2 year and 20 year maps and that wind anomalies are shown only for 0.5 and 1.0 m/s intervals.

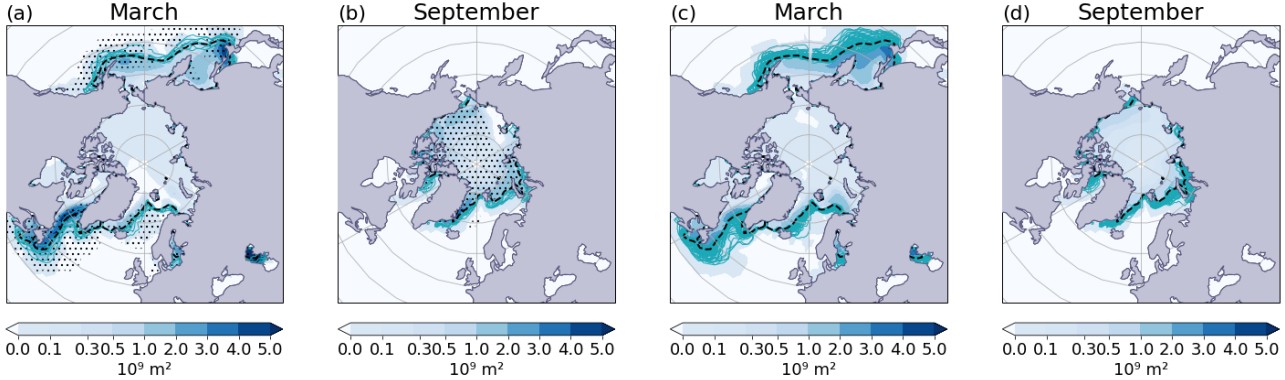

**Figure 4.** The March and September mean sea-ice area anomaly in contours and the sea-ice extent represented by the teal lines for 2 years after the eruptions (a) and (b) and 20 years after the eruptions (c) and (d). All variables are given as anomalies wrt 1-1850 CE. The black hashed line is the mean sea-ice extent in the control run and the $2\sigma$ significant sea-ice area anomalies are stippled.

The sea-ice extent as a response to the eruptions (Fig. 4a-b) is given for the maximum and minimum sea-ice area months, March and September respectively. For the short term response, the Arctic sea-ice extends southward by at least ~1-2° from the long term mean (past2k run 0), with the largest changes south of Greenland to south of Newfoundland and in the Pacific during March and in the Barents Sea and Kara Sea during September.

The long term response is shown in the right panel of Figs. 3 and 4. In both winter and summer the patterns are very similar to the 2 year response, only weaker for all variables but the sea-ice extent. For temperature, the main signal that remains is the cooling over the Hudson Bay, the Labrador Sea, east of Greenland towards Svalbard and east of Svalbard in boreal winter. In the same season, the largest SLP anomaly after 20 years is the increased pressure over Scandinavia and Siberia and the decrease over Greenland and the Mediterranean in boreal winter. The sea-ice extent for the long term is extended further south than for the first two years after the eruptions. Especially in the Labrador Sea in March and in lesser extent in the Barents Sea and Kara Sea in September, have an anomalous sea-ice extent, with up to ~5° and up to ~3-4° from the 1-1850 mean respectively.

Overall, the NH maps reveal that there is an atmospheric circulation change with a division at mid-latitudes (around 40°N to 45°N). In addition there is a land-sea contrast, with a larger cooling over land. The signal over ocean is opposite for precipitation and evaporation. These patterns lasts up to 20 years after the four large eruptions, where the long term response is only significant for 2 m air temperature.

**Ocean - sea-ice response**

Figure 5 shows the time evolution of the barotropic streamfunction (BTS) and the Atlantic meridional overturning circulation (AMOC) from the short (0-2 years), mid (3-10 years) and long (11-30 years) term response. The BTS corresponds to the horizontal surface flow, and the AMOC corresponds to the vertical mass streamflow in the ocean.

In the first 5 years after the eruptions, a decreased transport is visible in the BTS from the American east coast all the way to the eastern basin of the North Atlantic of up to 3 Sv (Fig. 5a and b). From Fig. 5b and c it can be seen that 3-10 years after the eruptions, the subpolar gyre south of Greenland gets more confined to its center, as the flow strengthens in the center and weakens on its eastern and western sides. At the same time, the transport around 30°- 40°N switches from a weakening to a strengthening flow of more than 3 Sv. After 10 years, the signal in the BTS becomes very small and after 21-30 years, it is returns to around 0 anomaly. The only signal that lingers is the weakened eastern part of the subpolar gyre, although it does not pass the significance test. The right panels in Fig. 5 reveal a significantly decreased overturning circulation near the surface around 60°N between 2-5 years after the eruptions. From 3-5 years after the eruption, there is an increase in the streamflow visible at ∼2000 m depth, corresponding to a strengthening of the AMOC. This strengthening peaks 6-10 years after the eruptions with a significant increase of more than 1 Sv, and decreases again after 10 years. After 20 years, the strengthening of the AMOC has disappeared and the ocean circulation shows hardly any anomalies. Only the decrease and increase in AMOC after 2-5 years and 6-10 years, respectively, is significant for this volcanic model ensemble. This indicates that the long-term AMOC signal is overwhelmed by internal variability.

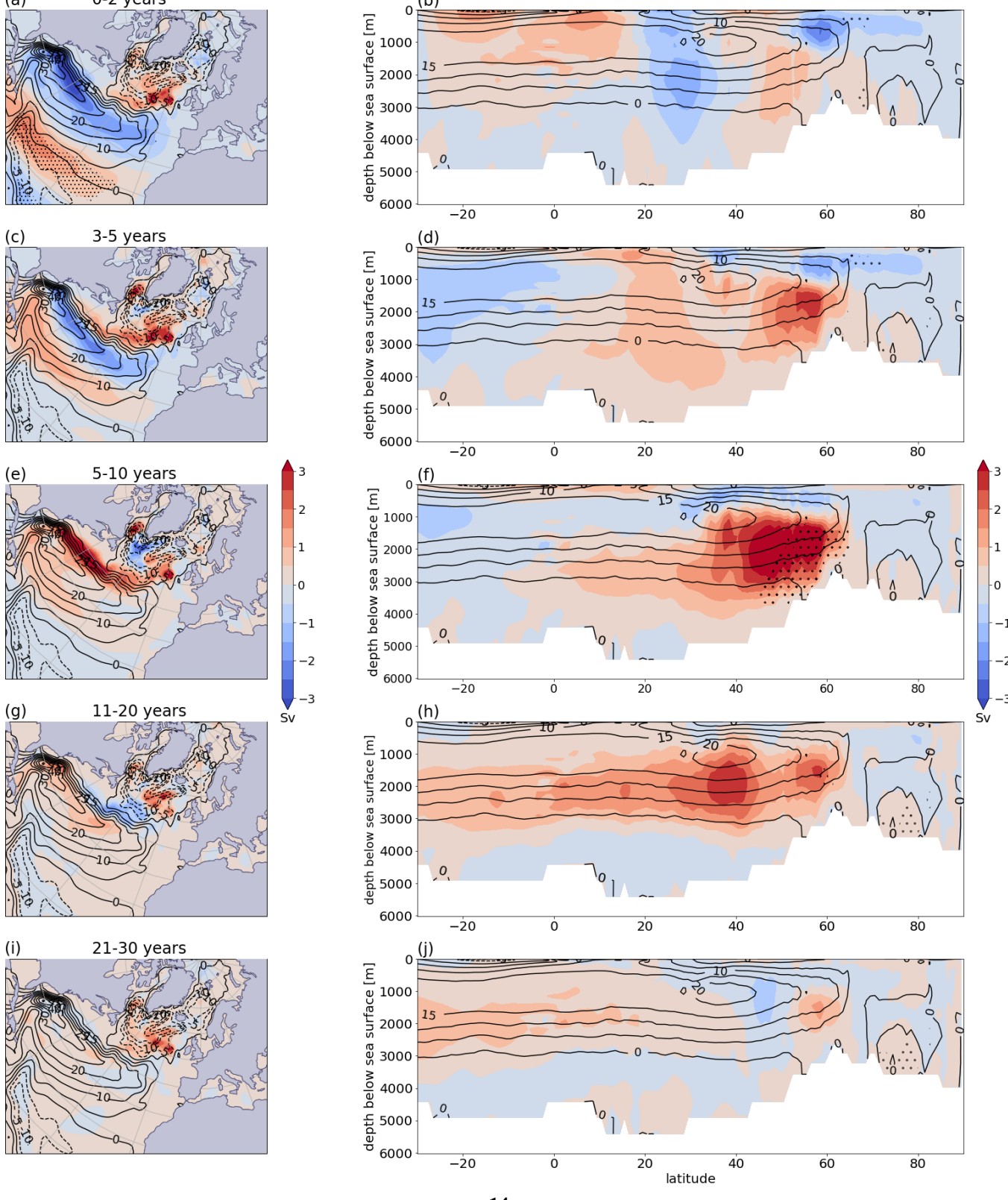

**Figure 5.**

**Figure 5**: (Previous page) Maps of barotropic streamfunction (BTS) and Atlantic meridional overturning circulation (AMOC) ensemble mean anomalies for 2 years (a, b), 3-5 years (c, d), 6-10 years (e, f), 11-20 years (g, h), and 21-30 years (i, j) after the four large eruptions. 1-1850 CE climatology in contours. For the BTS anomaly (left panels), negative absolute values indicate anti-clockwise rotation and positive absolute values indicate clockwise rotation. For AMOC (right panels), negative anomalies correspond to a decrease in stream flow, positive anomalies correspond to an increase in stream flow. Anomalies are with respect to the 1-1850 CE background climatology. $2\sigma$ significant areas are stippled.

## Model - tree ring comparison

Here, we compare the simulated 2 m air temperature anomalies to ensemble reconstructions from tree ring data from Büntgen et al. (2021), the NH1 reconstruction from Stoffel et al. (2015), and the N-VOLCv2 reconstruction from Guillet et al. (2020). For the model - tree ring comparison, the model temperature anomalies were taken for NH (40-75°N) land only during JJA, to correspond as closely to the tree ring data as possible.

In general, there is a good agreement between the 520-680 CE simulations and the tree ring data for the NH (Fig. 6). The reconstructed NH temperatures from the Büntgen et al. (2021) ensemble mean fall within the spread of the model simulations, except for a few years after the 540 CE eruption. The peak cooling for the model simulations is larger than the tree ring data, with a maximum cooling of more than 2.0 K for the model, 1.2 K for the Büntgen et al. (2021) tree ring ensemble mean, and 1.6 K for the Stoffel et al. (2015) and Guillet et al. (2020) reconstructions after the 536 CE eruption.

The temperature recovery is different for the Büntgen et al. (2021) tree ring ensemble and the model ensemble (Fig. 6). After the 536/540 CE volcanic double event, the modeled temperature recovers faster than the reconstructed Büntgen et al. (2021) mean temperature, in agreement with Stoffel et al. (2015) and Guillet et al. (2020). However, even though the Büntgen et al. (2021) ensemble mean falls outside the model ensemble range after 536/540 CE, some of the members from the reconstructions are still within the model ensemble spread. The model and tree ring data comparison shows a quite diverse picture dependent on the time period and reconstruction considered. In the three decades after the 536/540 CE volcanic double event, from around 545 CE to 575 CE, the simulated temperature anomalies are in good agreement with Stoffel et al. (2015) and Guillet et al. (2020), but smaller in comparison to Büntgen et al. (2021). Around 640 CE the model results and the reconstructed Büntgen et al. (2021) data agree quite well, while the Stoffel et al. (2015) and Guillet et al. (2020) data show a distinct minimum.

Another discrepancy between the model and the tree ring reconstructions is for small NH extratropical eruptions. In all the temperature reconstructions these eruptions give a small but distinct cooling peak. However, in the model mean they are not visible. The tree ring reconstructions from Büntgen et al. (2021) do, however, fall within the spread of the model ensemble, and for Stoffel et al. (2015) and Guillet et al. (2020) this is the case for the majority of the variability in the tree ring reconstructions. We will discuss the reason for the discrepancies in the discussion section.

Lastly, the timing of the peak cooling after the four large volcanic eruptions are in agreement. Except for the timing of the 626 CE eruption, which is shifted by one year. This is due to unknown eruptions set to January in the model forcing, while documentary evidence suggests the eruption took place in autumn (Sigl et al., 2015). Thus, the trees reacted the summer after the eruption in 627 CE, whereas they respond in the model simulations already in the summer of 626 CE.

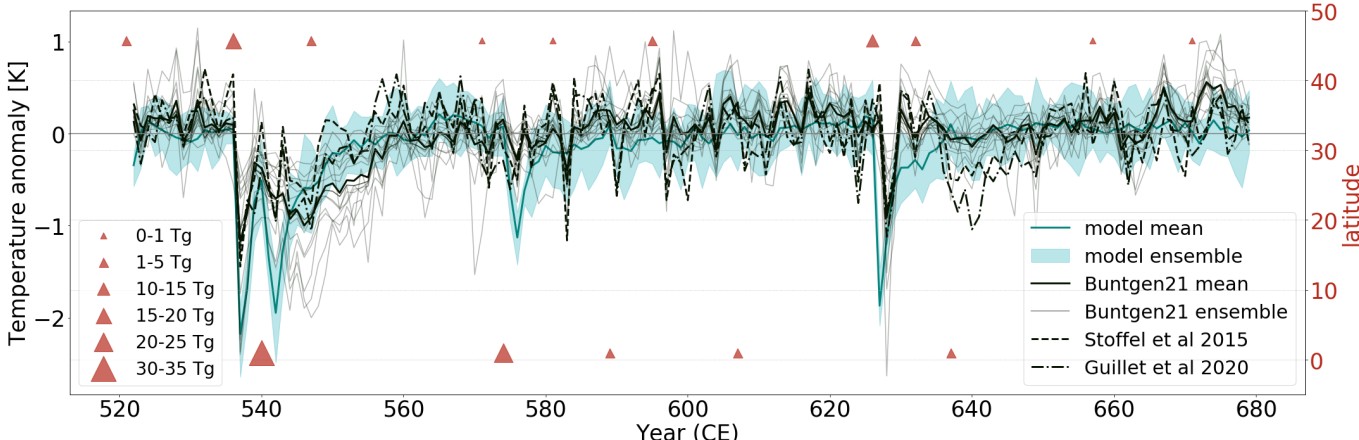

**Figure 6.** Model–tree ring comparison for the NH. The model 2 m air temperature anomalies are taken for land only, JJA and 40 to 75 °N. Climate model mean and the spread of the model ensemble; tree ring data for NH1 (Stoffel et al., 2015), N-VOLCv2 NH reconstruction (Guillet et al., 2020), and the mean and the ensemble of Büntgen et al. (2021). Anomalies are wrt 1-1850 CE except for the Guillet et al data which are wrt 500-1850 CE.

## 4    Discussion

**Volcanic response**

There is a strong cooling visible around the mid-latitudes in Fig. 3a, which corresponds to the peak accumulated AOD in the volcanic forcing (Fig. 1). The volcanic forcing used for this study is based on eVolv2k (Toohey and Sigl, 2017, see section 2.1.4), which gives the peak AOD at a latitude of 45°N (Fig. A5). The accumulated AOD is calculated for 15 years, to compare to Toohey et al. (2016a). The model reconstruction for the 536/540 eruption double event (Toohey et al., 2016a, see also Table 1) led to the maximum accumulated AOD in the Arctic at around 75°N. The simulated maximum cooling in mid latitudes at around 30-40°N in the Mediterranean, the Himalayas and the Western United States that we see in this study are the same areas as Toohey et al. (2016a) simulated. However, the cooling in our study is stronger, which could be due to the latitudinal difference of the peak volcanic forcing between the two studies. The fact that the difference in latitude for the peak AOD gives the same cooling patterns points to this not being from the direct effects of the volcanic aerosols.

**Atmospheric circulation changes**

In boreal winter, there is a see-saw pattern visible in the 2 year SLP response with decreased pressure over Greenland and an increase in pressure over Northern Europe, corresponding to the seesaw winter temperature pattern observed between Greenland and Scandinavia, as described by Van Loon and Rogers (1978). The hemispheric changes in boreal winter reflect a positive Arctic Oscillation pattern, as described by Thompson and Wallace (1998), which are typically observed after volcanic eruptions (Robock and Mao, 1992; Stenchikov et al., 2006).

After the volcanic eruptions, the climatological high pressure center over the Atlantic in the mid-latitudes gets more con-
fined, which leads to a separation at ∼45°N in the surface climate patterns. From Fig. 3, the 2 m air temperature, precipitation
and SLP anomalies do not show a clear positive NAO phase for the first two years after the four large eruptions, but from Fig.
A4, it can be seen that the ensemble simulates a positive NAO response in the first year after three of the four large eruptions.
There is a very small, non-significant warming over western Scandinavia and northwest Siberia in the DJF 2 m air temperature
anomaly, hinting towards a regional surface winter warming. Bittner et al. (2016a) analyzed a 100 member ensemble to study
the polar vortex and NAO response after volcanic eruptions in the tropics, and they concluded that for large eruptions (Kraka-
toa/Pinatubo size) at least 15 ensemble members are necessary to get a significant response on the polar vortex. A surface
winter warming pattern and a positive NAO has been observed after large tropical eruptions in the past (Robock and Mao,
1992; Robock, 2000). However, not all IPCC models show this signal (Stenchikov et al., 2002; Driscoll et al., 2012). The cause
for this is still highly debated. Some explanations for this deficiency include low top versus high top models (Charlton-Perez
et al., 2013), volcanic aerosol forcing details (Toohey et al., 2014), strength of the volcanic eruption and forcing (Bittner et al.,
2016b), tropical versus high latitude eruption impact (Schneider et al., 2009), the role of the ENSO state during the eruption
(Coupe and Robock, 2021), whether or not the observed signal is due to volcanic eruptions at all (Polvani et al., 2019). Here,
we use a mean of 12 ensemble members, 4 eruptions, and a mix of tropical and extratropical eruptions, which may explain the
lack of a significant NAO response.

The SLP anomaly in the long term response over the Barents Sea and Kara Sea corresponds to the sea-ice extent in Septem-
ber. The increase in sea-ice leads to lower air temperature above this area, which causes the air to descend, increasing the SLP.
The extended sea-ice in Labrador Sea is important for ocean and sea-ice interactions, which impacts the ocean circulation in
the North Atlantic (Zhong et al., 2011; Moreno-Chamarro, 2016).

**Hydrological cycle changes**

The increase in precipitation over the Mediterranean in boreal summer 2 years after the eruptions in the model simulations in
this study are related to the shifting of the inter tropical convergence zone (ITCZ) into the Southern Hemisphere (SH) after the
eruptions (not shown here), as well as a weakening of the high and low SLP over the North Atlantic (Fig. 3b). After a large
volcanic eruption, the ITCZ shifts away from the hemisphere that experiences the strongest cooling (Schneider et al., 2009), in
this case the NH. The southward shift in the ITCZ leads to a weakening of the northern branch of the Hadley cell (Wegmann
et al., 2014), which leads to a lower high pressure over southern Europe. This in turn leads to a wetter Mediterranean region,
especially in summer (3b-c). Another pattern that becomes evident in the boreal summer months is the land-sea contrast. Over
the continents there is a high pressure anomaly, whereas over the oceans there is a low pressure anomaly. This could be due to
the radiative effect, where the land responds faster to the cooling effect of the volcanic aerosols.

In addition, the evaporation is increased where the precipitation is also increased, and vice versa. This indicates that the
evaporation is more driven by soil moisture availability than by temperature changes. In a colder atmosphere, the air cannot
hold as much moisture and becomes saturated sooner, limiting the evaporation (Bala et al., 2008). If the evaporation is driven

by the change in temperature one would expect the evaporation to decrease with decreasing 2 m air temperature. In this case, there is an increase in precipitation over areas with the strongest cooling, like the Mediterranean, leading to more water being available for evaporation. North of 45°N over the continents, the evaporation anomaly decreases, which corresponds to the decrease in precipitation.

The simulated wettening over the Mediterranean and the drying over Northern Europe in boreal summer is consistent with other studies.Xoplaki et al. (2021) reconstructed the hydroclimate from speleothems and lake sediments for the eastern Mediterranean, and found a sharp change to wetter conditions in the second half of the 6th century for all sites. Iles et al. (2013) used HadCM3 and concluded the mean response to 18 eruptions during 1442-1992 to be a wettening over the Mediterranean and a drying of Northern Europe for up to 4 years after the eruptions during the summer season. Interestingly, our results on hydroclimate and SLP are in accordance with Liu et al. (2020), who used the Community Earth System Model (CESM) to simulate the volcanic effect of the Samalas 1258 CE eruption on the surface climate. In their study, they concluded that the dipole between the drying over Northern Europe and the wettening of the Mediterranean is a result of the weakening of the pressure systems over the North Atlantic and the resulting wind anomalies, as we found in this study for the 521-680 CE ensemble as well. Fischer et al. (2007) carried out composite analysis on multi-proxy data and found a wettening over the Mediterranean the year of eruption, which disappeared 1 year after, and using proxy data, Rao et al. (2017) found a wettening over the Mediterranean lasting up to 5 years, which was significant in the western Mediterranean. The low significance for the precipitation response could be due to the large internal variability, which drowns out the volcanic signal (Fischer et al., 2007). Helama et al. (2021) describe an East Atlantic pattern during the study period, corresponding to clear sky conditions over Northern Scandinavia, as obtained from tree ring proxies (TRW, MXD, and stable carbon isotope ($\delta$13C)). This pattern of reduced cloudiness is consistent with the dry conditions simulated over Northern Europe in our model simulations.

Hardly any studies have studied the long term response for hydrology after volcanic eruptions. Stevenson et al. (2018) argue that volcanic influences on the multidecadal hydroclimate variability is connected to the Atlantic multidecadal oscillation (AMO) teleconnections, which relates to our results of the SLP anomalies and the corresponding hydroclimate anomaly patterns. However, their focus is on tropical regions and not on the NH mid- to high latitudes. In our simulations, the precipitation and evaporation long term response show the same pattern as for two years after the eruptions, but this is not significant.

**Ocean - sea-ice response**

Zhong et al. (2011), and Miller et al. (2012) argued that the ocean - sea-ice feedback could play a major role in sustaining a century long cooling after a cluster of four volcanic eruptions in the mid 13th century. In contrast to these studies, our simulated cooling is shorter and lasts for several decades. After the 536/540 CE volcanic double event, the ensemble mean sea-ice cover does not reach the climatological mean value until the year 560 CE. This means that the sea-ice is in an anomalous state for more than 20 years after the 536 CE eruption. Following the 574 eruption, the sea-ice area anomalies persist for 30 years, and

after the 626 eruption for 15 years.

The ocean - sea-ice feedback can help to maintain the long-lasting cooling in the NH. A complex interaction between sea-ice expansion, changes in horizontal circulation (BTS) and overturning circulation leads to the reduction in heat transport, which then further enhances sea-ice expansion (Zhong et al., 2011). In our simulations, the sea-ice expansion is most pronounced in the Labrador Sea, which results in a confinement of the subpolar gyre to its center south of Greenland, where it shows a strengthening of ∼3 Sv 2-10 years after the eruptions (5b and c). At the same time, the cyclonic circulation weakens by more than 3 Sv in the eastern basin, which goes along with a reduction of the gyre related heat transports, as described by Jungclaus et al. (2014). In addition, there is a reduction in the Gulf Stream of more than 3 Sv in the first 5 years, followed by a strengthening of the same magnitude in years 6-10. This is consistent with Zanchettin et al. (2012), although they found the strengthening to occur 11-15 years after the eruption.

In the first 5 years after the eruption, we diagnose a decreased transport in the BTS from the American east coast to the eastern Basin of the North Atlantic. This decrease reflects a weakening of the anticyclonic subtropical gyre (STG), which is most pronounced in the Gulf Stream region. Positive BTS anomalies at the eastern side of the cyclonic subpolar gyre (SPG) indicate a weakening of the gyre circulation there, whereas negative anomalies south of Greenland point to a strengthening of the gyre center. The weaker gyre circulation in the eatsern basin can be associated with less gyre-driven heat transports at mid-latitudes (Moreno-Chamarro et al., 2017). After 5 years, the STG anomalies reverse, probably due to changes in the wind forcing, whereas the SPG anomalies in the gyre center and its eastern side are retained. We note, however, that these changes do not pass the significance criterion and we assume that therefore, the changes in the heat transport at 60°N (which should be largely gyre-driven) are consistent with the increased sea-ice extent, but not statistically significant.

As for the overturning circulation, the expanded sea-ice in the Labrador Sea leads to a deep water formation outside the normal deep water formation areas further south. The sea-ice formation process includes brine rejection, which leads to saltier, and thus more dense surface waters (Zhong et al., 2011), initiating overturning and thus leading to the deep water formation and an increased AMOC from 5 to up to 20 years after the eruptions in this study. The increased AMOC then in turn corresponds to the positive anomaly in the subtropics to mid-latitudes of the BTS. Halloran et al. (2020), concluded by analyzing oxygen isotope variability recorded from Iceland and the PMIP3 last millennium ensemble, that the same feedback cycle took place in the pre-industrial millennium. In addition, they demonstrate that a third of the multidecadal Arctic sea-ice variability can be explained by natural external forcing.

Zanchettin et al. (2012) simulated the ocean-atmosphere response to large volcanic eruptions, and concluded that the sea-ice extent and corresponding deep ocean convection in the North Atlantic was dependent on the initial state of the ensemble member. This led to different model ensembles having different spatial patterns when it comes to deep convection in the North Atlantic. The study from Zhong et al. (2011) about the onset of the LIA also concluded the response to be dependent on the initial state of the North Atlantic, as only 2 out of 4 simulations (one warm and one normal NA state) lead to a cooling long enough to resemble the LIA. It was part of our experiment design to start the ensemble simulation from an ocean state repre-

senting 520 CE conditions and to create the model's ensemble spread by perturbing the atmosphere. While this was done to include a proper representation of the forcing history in the previous decades, it does not allow us to investigate the effects of different ocean initial conditions (for this, an ensemble of past2k simulation would have been needed). It is therefore possible that the ensemble spread is underestimated. Another choice would have been to start the model integration from different states

of the AMOC (Pohlmann et al., 2004). An inspection of the AMOC time series, however, (Fig. A4 d) reveals that the AMOC variations do diverge quite rapidly after the start of the experiment and show a similar range at the time of the first major eruption compared to the end of the experiment. The only quantity where we see an increase of ensemble spread throughout the experiment is the global ocean heat content.

Another reason for the different responses between the studies described above, besides the different models used and the difference in initial conditions, could be the difference in volcanic forcing. The climate simulations are sensitive to both the climate model and the volcanic forcing, so using different ones will give a different climate response. This will be the more likely reason for the lack of consensus, as 10 ensemble members were run, which showed a range of variability in the ocean variables that are in the same range as the 1-1850 CE variability, and the response to the volcanic eruptions clearly stood out

against this, see for example Fig. 2. Other studies (Zanchettin et al., 2012; Zhong et al., 2011; Lehner et al., 2013) used a range of different background conditions in the beginning of their model simulations and found similar results as we did, despite the background conditions set-up. Zanchettin et al. (2016) showed that there are large uncertainties between different volcanic forcing sets that have been used for different modeling studies. Previously used forcings, like for example Gao et al. (2008) or Crowley and Unterman (2013), have a range of very different AOD evolutions over time after an eruption. Some forcings give

a very high peak but a short lifetime, while others have a longer lifetime with a lower and later AOD peak. In addition, the eruption season is also important, as different eruption seasons give different atmospheric circulation patterns and therefore, influence the transport of the sulfate aerosol, leading to different surface responses (Toohey et al., 2011). These different forcing patterns will give different responses of the surface climate in an ESM simulation. The volcanic forcing used in this study is the same as for PMIP4, which makes it easier to compare different model runs for this period. Further research into the role

of different forcing sets and different responses between the models is needed to fully understand this lack of consensus in the response timing between the studies. This could be a task within for example the Volcanic Forcing Model Intercomparison Project (VolMIP, Zanchettin et al., 2016).

**Model - tree ring comparison**

The model - tree ring comparison with the Büntgen et al. (2021) tree ring ensemble reconstruction (Fig. 6) shows a very good

agreement in the timing of the peak cooling of the 2 m air temperature anomalies for the land only extratropics of the NH during JJA. The mismatches that are still present in this NH comparison, like the strength of the peak cooling, as well as the lag after the 536/540 CE eruptions, include potential deficiencies and uncertainties regarding the method. For example, for the Büntgen et al. (2021) tree ring reconstructions TRW was used, which is known to give a lagged and smoothed response (Esper et al., 2015; Zhang et al., 2015; Lücke et al., 2019; Zhu et al., 2020). This could explain the lag after the volcanic double event

for Büntgen et al. (2021), whereas the other two reconstructions, Stoffel et al. (2015) and Guillet et al. (2020), are more in line with the model simulations. As reconstructions become more uncertain further back in time due to the sparseness of available proxy records, which mainly rely on tree ring records. This is the case especially before 1200 CE (Masson-Delmotte, 2013; Esper et al., 2018; Neukom et al., 2019), and therefore, we have chosen to use the Büntgen et al. (2021) reconstruction as it uses the same number of tree ring records throughout the entire CE. Additionally, we have chosen to include the reconstructions by

Stoffel et al. (2015) and Guillet et al. (2020) as they both consist of a mix of TRW and MXD records. Testing a comparison of the model results with a multiproxy temperature reconstructions for the entire NH annual mean (Neukom et al., 2019) reveals a worse agreement (see Appendix A, Fig. A3).

Timmreck et al. (2021) showed that beyond model deficiencies, choices regarding both volcanic forcing strength and spatial

structure can similarly affect reconstruction–simulation comparisons. Hence, a potential reason for the offset between the model and the tree ring records after the 536/540 CE eruptions could be that the forcing of the moderate eruption in 547 CE is underestimated. Other smaller NH extratropical eruptions also have a weaker temperature signal in the modeled temperature response, which could point to an under-representation of the volcanic forcing of small to medium size eruptions.

The EVA forcing generator that was used to get the AOD input for the model (see methods) was based on tropical eruptions

(Toohey et al., 2016b) and might therefore not simulate the processes and related timescales for relatively small eruptions in the extratropics. The fact that the eruption dates are always put to January if unknown also could influence the simulated response (Toohey et al., 2011; Stevenson et al., 2017; Toohey et al., 2019).

The concept of a LALIA was raised by Büntgen et al. (2016), based on tree ring data from the Alps and Altai. Comparing

these specific sites, the model and tree ring reconstruction do not agree as well as they do for the NH tree ring reconstructions, and especially the Altai reconstruction shows a longer cooling after the 536/540 CE eruptions (Fig. A2b and c). In addition, Matskovsky and Helama (2014) and Helama et al. (2021) report a century long cooling for Northern Scandinavia from 530 to 650 CE based on MXD, TRW, and $\delta13C$ data, which is not supported by our model simulations (Fig. A2a). Other proxy-based studies (Larsen et al., 2008; Esper et al., 2012b; Luterbacher et al., 2016; Helama et al., 2017; Neukom et al., 2019) found

a cooling up to 570 CE for Europe, Scandinavia, and the NH, based on tree ring, ice-core, lake sediment, and documentary records (Figs. A2a and d, and A3). Comparisons for the different tree ring data sets have been carried out by previous studies for the last millennium (Wilson et al., 2016; Lücke et al., 2019). However, not enough records exist for the first millennium to carry out a similar comparison for this period yet.

From the perspective of our model results, the persistence of the cooling was not as long lasting as the tree ring sites from

the Alps, Altai, and Northern Scandinavia suggest. Perhaps the century-long lasting cooling may be apparent in the Alps and Altai tree ring records, as a regional feature occurring at high altitudes of the mid-latitudes. Another possibility is, that our model resolution is too coarse to fully capture the topography of the Alps, Altai, and Northern Scandinavia or that the evolution of glaciers, which is not accounted for in the model, could explain why the model simulations deviate in these areas. Glacier fluctuations from the last 2000 years show glaciers in the Alps advancing during this period, lasting for $\sim$ a

century (Solomina et al., 2016). Ice core records from Greenland (Sigl et al., 2015) agree well with the tree ring data from Alps (Büntgen et al., 2016). The LIA also had regional variability and was also punctuated with warmer periods (Mann, 2002). The regional variability of the LALIA is a topic to be further investigated in future studies. Furthermore, additional tree ring density records from the 1st millennium, in particular from poorly replicated regions, are needed to obtain a clear picture of the mid-6th century long term cooling event.

Moreover, the study from Büntgen et al. (2021) shows that it is important to also use an ensemble when it comes to tree-ring reconstructions, as the different statistical methods used to analyze the data give different results. Even though the ensemble mean shows a discrepancy with the model simulation after the 536/540 CE eruptions, some ensemble members fall within the range of the model ensemble spread. Overall, the model ensemble shows less variability in particular to the 536/540 CE response but also to other volcanic eruptions than the tree-ring reconstruction ensemble. The reality can be viewed as one iteration of what could have happened under different initial conditions, ocean states and internal variability. Taking into account the entire range of ensemble members from both climate model simulations and proxy reconstructions is therefore important. In our follow up paper (van Dijk et al., 2022) the individual members are analyzed in more detail for Scandinavia.

Our model ensemble set up with the PMIP4 volcanic forcing reveals up to 25 years of surface cooling over land of the NH extratropics during summer and it also shows the distinct atmospheric circulation response pattern for precipitation and evaporation at $\sim$40°N, separating the Alps and Altai from the other tree ring locations. The change in hydro-climate corresponds to the soil moisture availability for the trees and thus could have impacted tree ring growth (Bassett, 1964; Müller et al., 2016). Future studies are needed to further investigate the temperature, atmospheric circulation, and hydro-climate response in a multi-model context and investigate regional differences in the tree ring records, as well as the response to tropical and extratropical eruptions.

## 5   Summary and Conclusions

In this study, we analyzed new climate model ensemble simulations from 520-680 CE with the MPI-ESM model and PMIP4 volcanic forcing. The aim was to get insights if a series of volcanic eruptions starting with the 536/540 CE volcanic double event would lead to a long lasting cooling in the mid 6th to 7th century in the Northern Hemisphere (NH).

In summary, we find that a series of four large and several small to medium size volcanic eruptions lead to a significant peak cooling up to 3 K in the NH. In addition, atmospheric circulation changes with a hemispheric dipole structure separated at around 40-45°N and a land-sea contrast pattern occurs. Analyzing underlying mechanisms in the North Atlantic reveal that complex interaction between sea-ice expansion, changes in the barotropic streamfunction and the Atlantic meridional overturning circulation lead to a reduction in the ocean heat transport, which then further enhances sea-ice expansion impacting NH surface climate for up to 20 years.

The modelled volcanic induced multi-decadal large scale NH cooling is consistent with tree ring reconstructions for this period. This shows that the ocean - sea-ice sustained $\sim$20 year cooling as occurring in our simulations is a plausible mecha-

nism. The simulated dipole in the atmospheric circulation during summer could hold clues for the difference in the tree ring reconstructions for the Alps and Altai sites, which show a century long cooling. Future model studies comparing different models and volcanic forcing with each other will be useful for verifying our results.


The summer cooling can have a serious effect on vegetation and society, as a few degrees cooling in summer can lead to crop failure and famine in areas that are close to the temperature limit for growing crops. In a follow-up study, we will investigate if and how large volcanic eruptions in the mid-6th century may have impacted climate and society in Scandinavia by using the model simulations from this study and combining it with a local model as well as climate proxy and archaeological records.

*Code and data availability.*  Primary data and scripts used in the analysis and other supplementary information that may be useful in reproducing the author's work are archived and can be obtained by request. Model results will be available under cera-www.dkrz.de. The PMIP4 past2k simulations contribute to CMIP6 and can be retrieved via the Earth System Grid Federation network (e.g., https://esgf-data.dkrz.de/search/cmip6-dkrz/). Tree ring proxy data can be obtained from NOAA/World Data Service for Paleoclimatology archive https://www.ncei.noaa.gov/access/paleo-search/study/33215.

*Author contributions.*  KK, CT and EvD conceived the original idea. EvD, CT, JJ and SL planned the model simulations. SL and JJ designed the model and the computational framework. EvD carried out the model runs, processed the data, performed the analysis, and designed the figures. KK, CT, JJ contributed to the interpretation of the results. KK supervised the project. All authors discussed the results and helped writing the manuscript.

*Competing interests.*  The authors declare they have no competing interest.

*Acknowledgements.*  This work is funded by the NFR/UiO Toppforsk project "VIKINGS" with the grant number 275191. Computations, analysis and model data storage were mainly performed on the computer of the Deutsches Klima Rechenzentrum (DKRZ), and partly on Sigma2 the National Infrastructure for High Performance Computing and Data Storage in Norway. CT acknowledges support to this research by the Deutsche Forschungsgemeinschaft Research Unit VolImpact (FOR2820,398006378) within the project VolClim (TI 344/2-1). Thanks to Ulf Büntgen, Michael Sigl, and Matthew Toohey for discussions on the results, and to Davide Zanchettin for his help with the significance calculations. Thanks to MPI for Meteorology Hamburg to make a guest exchange possible, which led to the idea for new model experiments of this paper.

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

## Appendix A - Model - tree ring, and multi-proxy reconstruction comparison

Most temperature reconstruction data sets go back to about 1200 CE, and the further back in time, the fewer proxy records remain, and the more uncertainties they contain (Masson-Delmotte, 2013; Esper et al., 2018; Neukom et al., 2019). The main proxy type that remains to reconstruct the temperatures in the Northern Hemisphere (and especially mid-high latitudes, Europe)

are tree rings (Neukom et al., 2019), and they are often used to reconstruct the temperature in especially Europe (Luterbacher et al., 2016). Other reconstructions available consist of a mix of proxies with a more limited dating precision, which leads to a reduction of the amplitude of the signals (Sigl et al., 2015; Büntgen et al., 2020; Plunkett et al., 2021). Only ∼25% of the proxies available for our study period have annual dating precision (Neukom et al., 2019).

The data sets used for the individual tree ring sites are from Northern Scandinavia (N-Scan) (Esper et al., 2012b, a), from the Alps and Altai (Büntgen et al., 2016) and from Europe (Luterbacher et al., 2016) (Table 2). We have also carried out a comparison for other individual sites in the NH (see tree ring sites in Fig. A1), but due to a high internal variability the volcanic

signal does not stand out for the model simulated temperatures. We could therefore not use these sites for a model - tree ring comparison.


**Table 2.** Overview of tree ring/proxy locations and type (MXD: maximum latewood density; TRW: tree ring width) used for comparison. For the proxy reconstructions from Stoffel et al. (2015) and Guillet et al. (2020) a nested reconstruction method was used, for the reconstructions from Luterbacher et al. (2016) and Neukom et al. (2019) the method used is composite plus scaling (CPS).

| Location name | Coordinates | Type of data | Reference |
|---|---|---|---|
| NH | 40°- 75°N, -180°- 180°E | TRW and MXD | Stoffel et al. (2015) |
| | | TRW and MXD | Guillet et al. (2020) |
| | | TRW | Büntgen et al. (2021) |
| NH | 0°- 90°N, -180°- 180°E | mixed proxies | Neukom et al. (2019) |
| N-Scan | 66°- 70°N, 19°- 29°E | MXD | Esper et al. (2012b, a) |
| Alps | 46°N, 12.5°E | TRW | Büntgen et al. (2016) |
| Altai | 50°N, 87.5°E | TRW | Büntgen et al. (2016) |
| Europe | 35°- 70°N, -25°- 40°E | TRW and MXD | Luterbacher et al. (2016) |

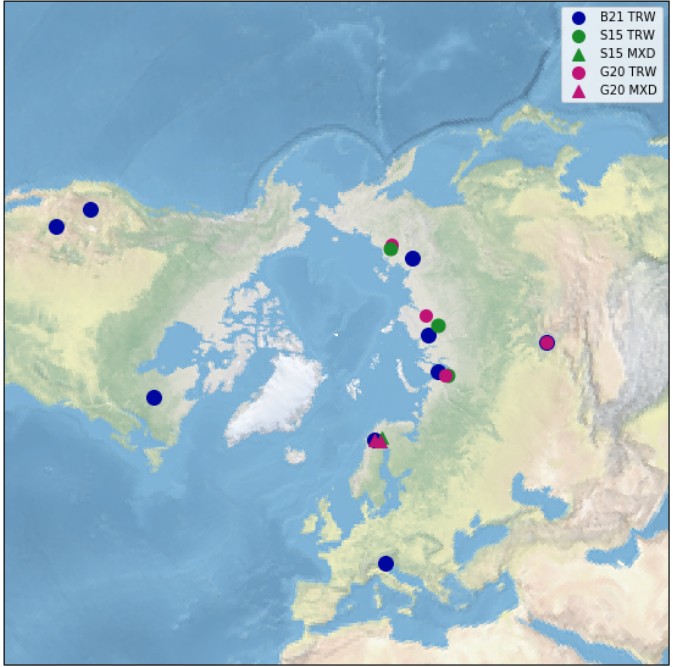

**Figure A1.** Locations of the (Stoffel et al., 2015, green dots), (Guillet et al., 2020, purple dots), and (Büntgen et al., 2021, blue dots) tree ring sites. For more information see also Table 2. Tree ring width data is represented by the circles and maximum latewood density by triangles.

The tree ring reconstructions for the Alps and Altai (Fig. A2b and c) show more discrepancies compared to the N-Scan site or the NH tree ring ensemble reconstruction. The tree ring temperatures still fall within the variability of the model simulations, but the maximum of the cooling for the four large eruptions does not always agree so well. For the Alps, the recovery time after the 536/540 CE eruptions is longer for the tree ring temperatures than for the model, showing a lag, as for the NH compilation.

For the Altai, the cooling in the tree ring reconstructions after the 536/540 CE eruptions lasts even longer until $\sim 660$ CE, revealing a large discrepancy between the model temperature and the reconstructed temperature for this site.

For Europe, the model and proxy data (Luterbacher et al., 2016) agree well. The proxy reconstruction falls within the spread of the model ensemble (Fig. A2d). As for the comparison with Büntgen et al. (2021), the peak cooling is less for the reconstructions, and there is a lag in the proxy data after 540 CE.

To illustrate the reduction in amplitude of the signals when using available multi-proxy reconstructions that represent the yearly mean, we compared the Pages2k reconstruction (Neukom et al., 2019) to the simulated NH annual mean near surface temperature (Fig. A3). The proxy data agrees well for the recovery period after the 536/540 CE volcanic double event, and from $\sim 650$ CE - 675 CE, but the reconstructions show a weaker cooling following the volcanic eruptions, and for the 574 CE eruption the reconstruction does not show a signal at all.

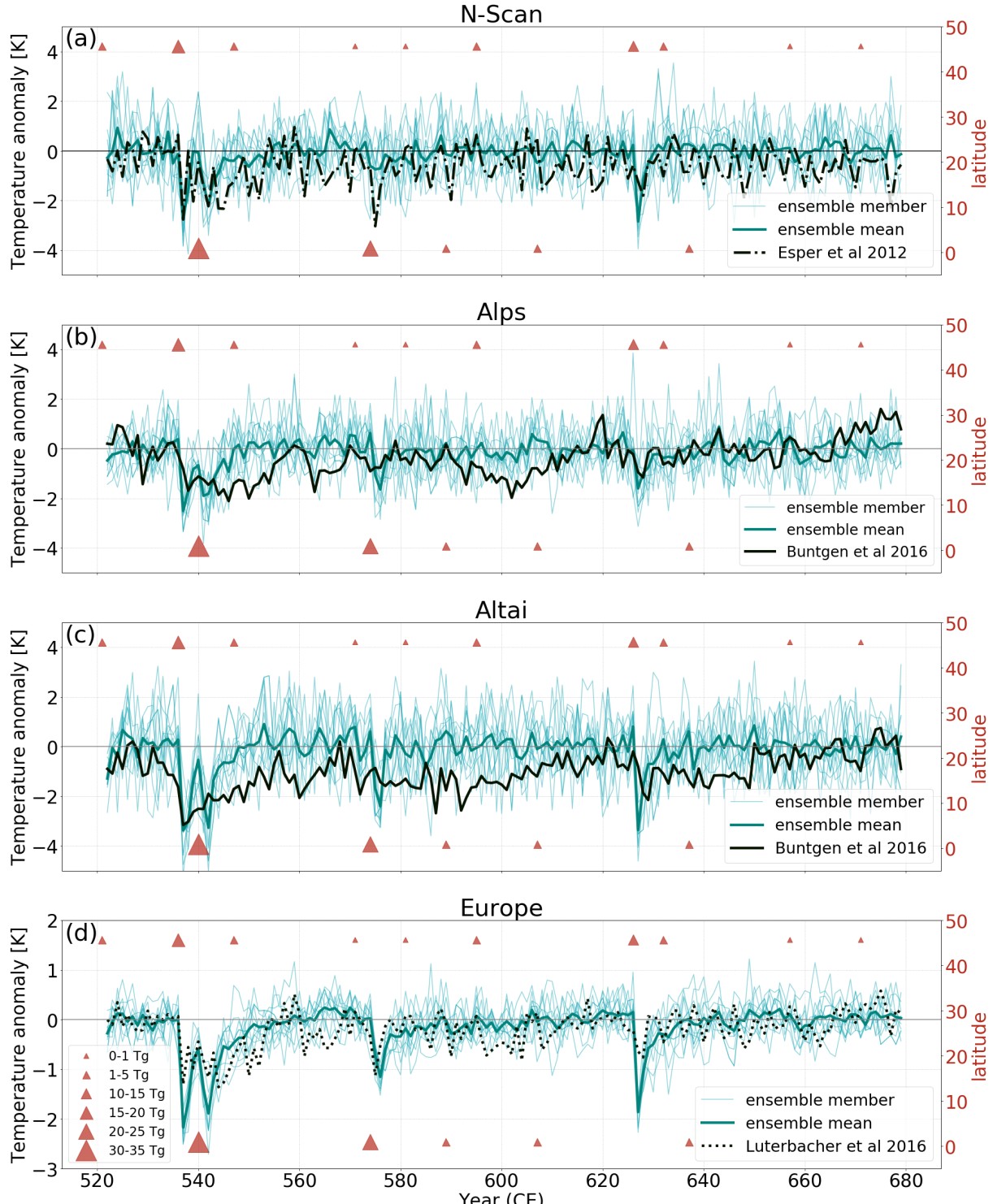

**Figure A2.** Model - tree ring comparison for a) Northern Scandinavia (N-Scan) (Esper et al., 2012b), b) the Alps and c) the Altai (Büntgen et al., 2016) and Europe ((Luterbacher et al., 2016). Mean anomalies are calculated for the 1-1850 CE reference period. The red triangles represent the volcanic eruptions from the model forcing according to their latitude (tropical or NH extratropical), and the size of the triangle corresponds to the estimated range of sulphur [Tg S] injected into the stratosphere (Toohey and Sigl, 2017).

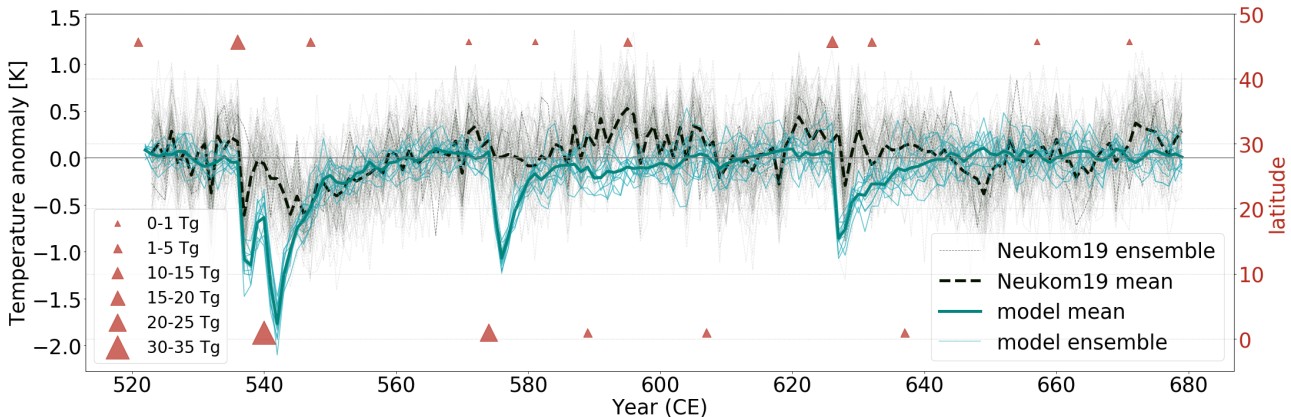

**Figure A3.** Model - proxy reconstruction comparison for the entire Northern Hemisphere with PAGES2k data (Neukom et al., 2019). Mean anomalies are calculated for the 1-1850 CE reference period. The red triangles represent the volcanic eruptions from the model forcing according to their latitude (tropical or NH extratropical), and the size of the triangle corresponds to the estimated range of sulphur [Tg S] injected into the stratosphere (Toohey and Sigl, 2017).

## Appendix B - NAO

The climate in the North Atlantic and Europe is often indicated by the state of the North Atlantic Oscillation (NAO). The NAO describes variations of the sea level pressure difference between Iceland and the Azores, leading to wind patterns carrying warm moist air to Northern Europe (positive NAO phase), or dry cold air (negative NAO phase). For this study, we calculated the NAO index using the method from Hurrell (1995). The NAO index is calculated by taking the first principle component of the SLP anomalies for the area 90°W to 40°E, 20°N to 80°N. The obtained time series then indicates whether the NAO is in a positive or negative phase, where a positive NAO corresponds to relatively warmer and wetter conditions over Scandinavia, and the negative phase corresponding to the opposite (Hurrell, 1995). See Fig. A4 and Table A1.

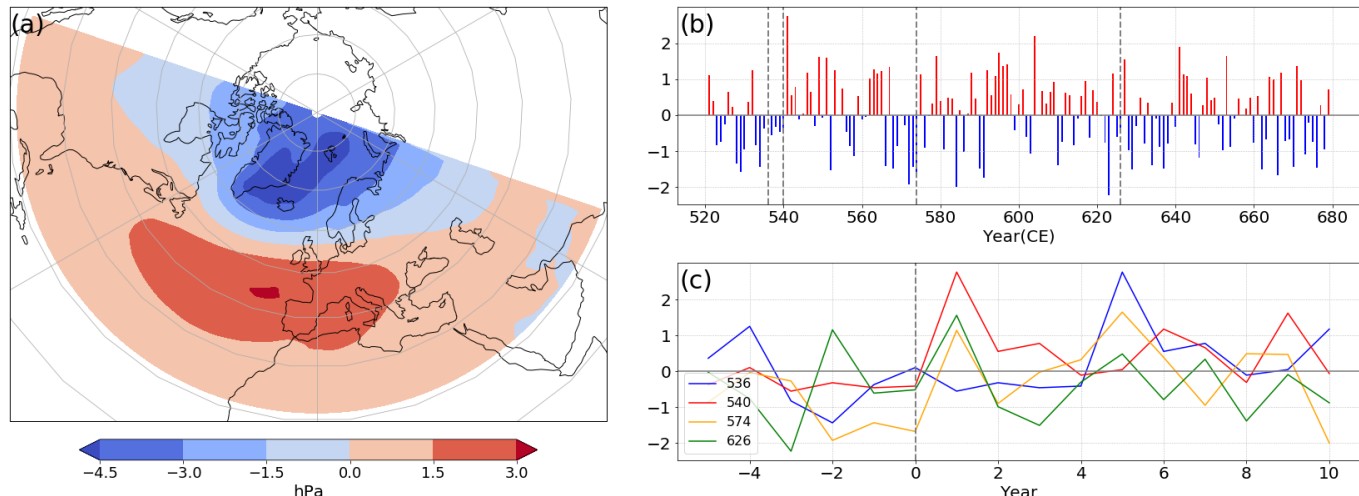

**Figure A4.** NAO DJF for (a) the climatological NAO signal for 1-1850 CE past2k "run 0", and the response as (b) a time series of the ensemble mean NAO index and (c) an epoch analysis of the NAO index for the four large eruptions during 520 to 680 CE. The eruption years are indicated with a dashed vertical black line in (b). DJF year 0 corresponds to December of the year before the eruption, and January to February of the year of the eruption.

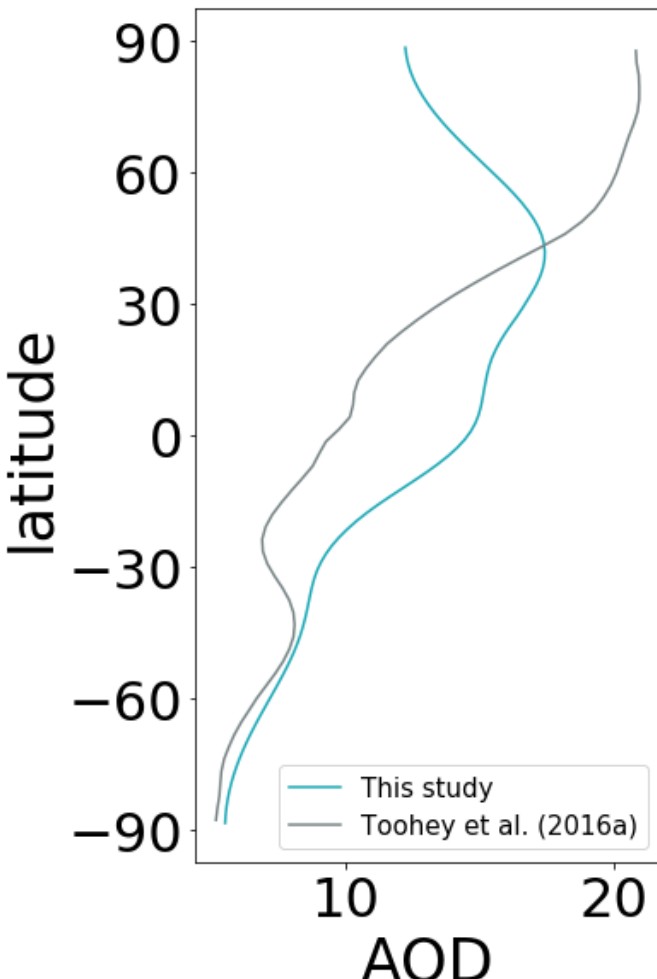

**Figure A5.** Comparison for accumulated AOD for 536-550 CE for the volcanic forcing from this study (eVolv2k(EVA), Toohey and Sigl, 2017) and from Toohey et al. (2016a) (their Fig. 2d, based on MAECHAM-HAM).

### Appendix C - Climate indices

Zanchettin et al. (2013) identified the importance of background conditions for ensemble-based numerical studies of large volcanic eruptions as they have an impact on the mechanisms involved in the post-eruption decadal evolution. To determine the background conditions, i.e., the initial climate state of our ensemble members and to address the climate variability during 520 CE to 680 CE several relevant climate indices were calculated.

**North Atlantic**

Since the focus of this study is on Europe, the initial state of the North Atlantic (NA) ocean and sea-ice was calculated. In Table A1, the Atlantic meridional overturning circulation (AMOC), the strength of the barotropic streamfunction (BTS) and the SST for the subpolar gyre are given; Fig. A6 displays the time series of these quantities for the 520-680 CE and past2k runs. The AMOC is defined as the maximum in the mass streamfunction, which occurs between 35°and 45°N and 800 m to 1200 m depth in the MPI-ESM past2k run 0 climatology (1-1850 CE mean). The column integrated BTS can be used as a measure for

the strength of the subpolar gyre, which is the horizontal flow south of Greenland with an anticlockwise rotation. The subpolar gyre in the climatology resides between 50°N and 65°N and 20°to 60°W, so we take this area as a measure for the state of the BTS and for the SST. From Table A1 and Fig. A6 it becomes clear that the initial ocean and sea-ice state before 536 CE covers mean North Atlantic conditions of the past 1850 years. See Fig. A6 and Table A1.

**ENSO**

The El Niño Southern Oscillation (ENSO) index was calculated using the Niño 3.4 index, as described by Trenberth and Hoar (1997). The sea surface temperature (SST) anomaly was calculated for the box 5°S - 5°N and 120°-170°W. If the 5-month running mean exceeds 0.4°C for 6 months or more, the ENSO state is defined as an El Niño (Trenberth and Hoar, 1997). La Niña is defined in the same way, but for -0.4°C instead.

Figure A6 and Table A1 give an overview of the ENSO conditions before the four large volcanic eruptions. The spread in

the ensembles cover a wide range of different ENSO states at the beginning of the transient runs.

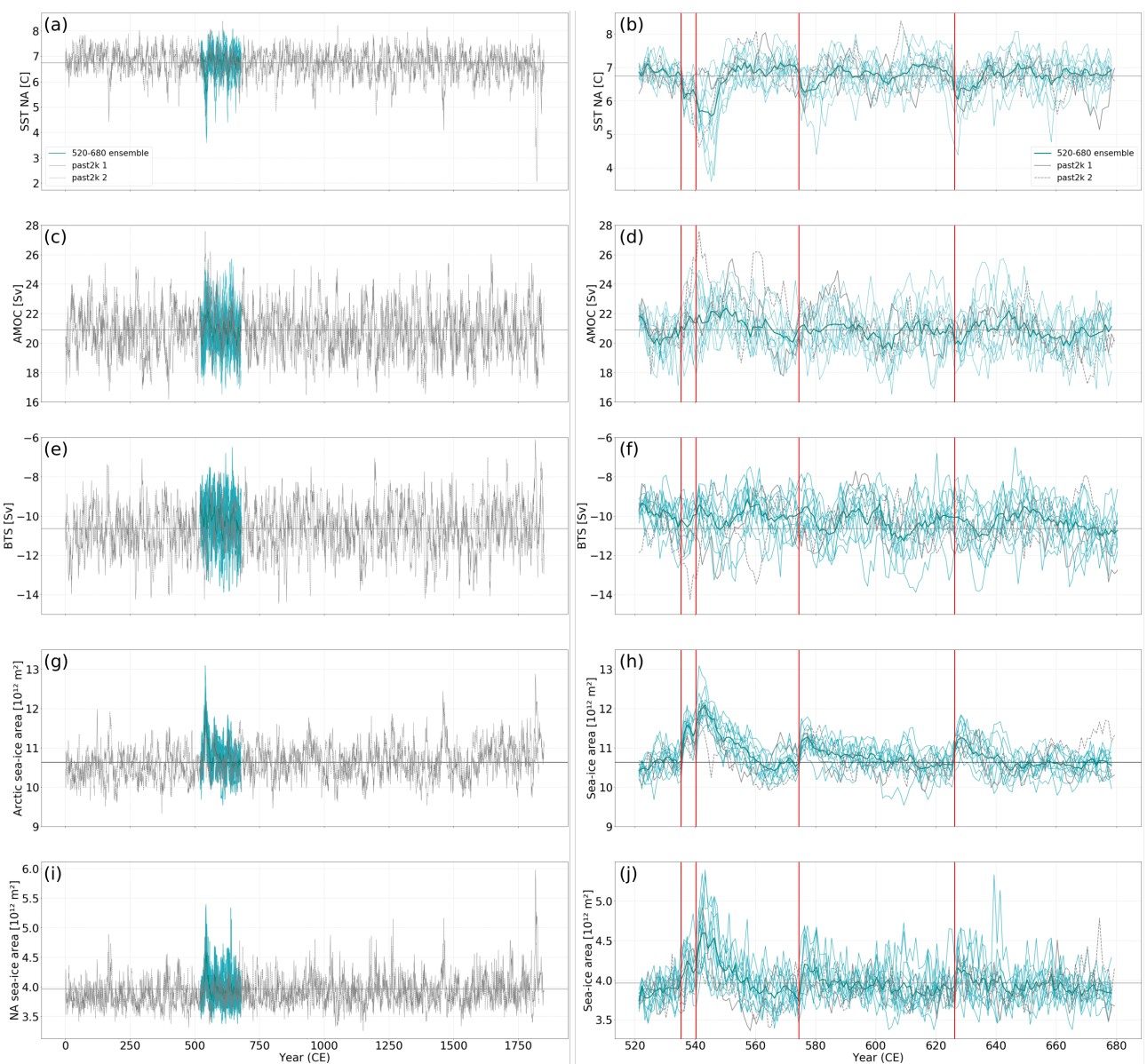

**Figure A6.** Time series for a/b) NA SST, c/d) AMOC, e/f) BTS, g/h) Arctic sea-ice area and i/j) NA sea-ice area for 1-1850 CE (left panels) and 521-680 CE (right panels). The red lines indicate the large volcanic eruptions in the study period, and the straight gray line is the mean of the past2k "run 0".

**Table A1.** Initial ocean and sea-ice conditions. Mean and [range] of all ensemble members for the AMOC, subpolar gyre BTS and SST, NA March sea-ice area, and ENSO states (Niño3.4 index) before the four large eruptions. *For all quantities the mean of the previous year of the eruptions are calculated except for the Niño3.4 index, where the 5 month rolling mean (ASOND) of the previous year was taken.

| Variable* / Eruption | 536 CE (extratropical) | 540 CE (tropical) | 574 CE (tropical) | 626 CE (extratropical) |
|---|---|---|---|---|
| AMOC [Sv] | 20.96 [19.16, 21.77] | 21.81 [18.76, 25.82] | 20.35 [18.53, 24.25] | 20.92 [19.73, 22.31] |
| Subpolar gyre BTS [Sv] | -10.35 [-8.75, -12.26] | -10.09 [-8.79, -11.29] | -10.16 [-8.71, -11.41] | -10.22 [-8.71, -12.76] |
| Sub-Polar Gyre SST [°C] | 6.69 [6.22, 7.16] | 6.28 [5.09, 6.82] | 6.97 [6.22, 7.69] | 6.65 [4.95, 7.59] |
| NA March sea-ice area [$10^{12}$ m$^2$] | 3.95 [3.57, 4.48] | 4.10 [3.68, 4.84] | 3.86 [3.54, 4.23] | 3.88 [3.65, 4.65] |
| Niño3.4 index | 2 neutral 5 La Niña 5 El Niño | 1 neutral 10 La Niña 1 El Niño | 7 neutral 1 La Niña 4 El Niño | 6 neutral 4 La Niña 2 El Niño |