# Peer review of "Was there a volcanic induced long lasting cooling over the Northern Hemisphere in the mid 6th-7th century?"

_Climate of the Past, 2021_

## Author Comment (AC1)

Dear editor and reviewers, thank you for the detailed review of the paper. We have compiled a point by point response going through all comments of the four reviews. Our comments are given in blue and bold text.

**General reply:**

We have considerably revised the manuscript by rewriting the abstract, introduction and summary and conclusion, as well as separating the discussion from the result section. We updated the references in the introduction and discussion. As suggested, we now use the Büntgen et al. (2021) tree-ring reconstruction ensemble for the model - tree-ring comparison (see figure below) and have rewritten the comparison section accordingly, focusing on the Northern Hemisphere temperature anomaly.

**Reviewer 1:**

Summary: The manuscript focuses on the climate response to the strong volcanic eruptions that occurred in the 6th century. The authors analyze an ensemble of climate simulations with the model MPI-ESM, compare their results with other previous simulations and with temperature reconstructions based on dendroclimatological data. The analysis of model data includes the atmospheric response (temperature, precipitation, wind) and the response of the ocean circulation in the North Atlantic. One important issue is the duration of the response of these variables, as previous simulations and decollimated reconstructions seem to disagree - at lest in some regions.

Recommendation: In my opinion the scope and focus of the article are valuable. There are some knowledge gaps about the response of large volcanic eruptions and the origins of Little Ice Ages in the last millennia, that need to be filled.

However, the manuscript leaves clear room for improvement, and I believe it requires considerable revisions. The structure is sometimes confusing, the language is also not always clear, and the data analysis is not deep enough. This is reflected in the unclear take-home messages of the manuscript. Disagreements between simulations are explained by general model differences. Disagreements between reconstructions and simulations are explained by possible errors in the volcanic forcing or deficiencies of the proxy data, mainly in the tree-ring data, but the manuscript does not include

more specific and solid explanations. Sometimes, the discussion is inconsistent, and some examples of this are given in the list of particular points below.

Thanks for your constructive comments. We have taken your and the three other reviewers' comments into account in the revised manuscript, as further answered above in the general comments and below in detail.

1. ' sea level pressure and a decrease in hydrological variables occur'

The text could be here clearer. Does this sentence mean that both precipitation and

evaporation become smaller?

The sentence indeed refers to both precipitation and evaporation. We have rewritten the abstract (see reviewers 2 and 3's comments) and we have taken your comment into account.

'...a decrease in both precipitation and evaporation poleward of....'

2. 'However, most reconstruction data sets go back to about 1200 CE, and the further back in time, the fewer proxy records remain, and the more uncertainties they contain (Masson-Delmotte,2013; Neukom et al., 2019). The sentence is not clearly formulated (perhaps it is not grammatically correct). Consider an alternative formulation like ' Further back in time the network of proxy data becomes sparser and uncertainties in each individual temperature record also grow.

We have revised the introduction due to reviewers 2 and 3's comments and now write the following:

"Multiproxy reconstructions in the NH during this period consist of mainly tree-ring, marine sediments, and ice core records. Marine sediments have a coarse chronological resolution and would therefore not capture the volcanic signal, and ice core records are confined to the Greenland Ice-sheet (PAGES 2k Consortium, 2017). Tree-ring records have an annual resolution with an absolute dating. Therefore, we use the most recent tree ring ensemble reconstruction for the past2k (Büntgen et al., 2021) to compare to the model simulations in this study."

We now use the tree rings ensemble data from Büntgen et al. (2021) as a reference data set, where one finds a detailed discussion about proxy uncertainties. This info has been added.

3. 'The aim of this study is to investigate whether a multidecadal to centennial cooling may have occurred in the mid 6th to 7th century.'

This sentence states the main objective of the study. However, the reader will not be wiser after reading the manuscript. The disagreement between the proxy records that do indicate a centennial cooling do not agree with the model results, but the study does not include a solid explanation for this disagreement (apart from speculating that perhaps the high altitude of the Alp records may be responsible for the long temperature recovery).

This is perhaps my main concern with the manuscript. It is in general too descriptive and does not go deep enough into explaining those disagreements.

According to the suggestions of reviewers 3 and 4, we now include the new tree-ring reconstruction ensemble from Büntgen et al (2021). Thereupon, the disagreement of proxy records and model data is not the focus of our study anymore but a motivation for our study. We also refer to the discussion in Timmreck et al (2021) about the uncertainties in the model simulations. Therefore, the aim of the study has not changed/ has changed to "The aim of this study is to investigate whether a series of volcanic eruptions induced a multidecadal to centennial cooling in the mid 6th to 7th century.".

4. ' the short term (years), as well as the long term (decadal to centennial)

the short term (annual)...

Thanks, corrected.

5. 'sea ice impacts, we also study atmospheric and ocean circulation, hydrology and the ocean-sea ice feedbacks in maintaining the climate signal'

the 'climate signal' is too unspecific. Please, help the reader by being more specific, for instance 'in maintaining the volcanic induced cooling'

Thanks for the suggestion, the sentence has been reformulated as follows:

'...we also study atmospheric and ocean circulation, hydrology and the ocean - sea-ice feedbacks in maintaining the volcanic induced surface cooling.'

6. 'For this study, we ran ten ensemble members for 160 years from 520-680 CE.'

for 160 years, starting in 520 CE. Or alternatively, covering the period 520-680 CE

The sentence has been reformulated as follows:

'For this study, we ran ten ensemble members, covering the period 520-680 CE.'

7. For each ensemble member the atmospheric diffusivity was changed by  $1 \cdot 10-5$  to simulate slightly different climate states by the year 536 CE, the year of the first large volcanic eruption.

physical units are missing

Thanks, we added the units:

'For each ensemble member the atmospheric diffusivity was changed by 1·10-5 m2s-1...'

8. Historical Land Use Data Set for the Holocene (HYDE3.2, Klein Goldewijk, 2016). Considering several options (e.g. linear ramp-up) we decided to simply let the land-cover data be constant for the first 850 years of the past2k runs.

We prescribed a constant land-cover for the first 850 years of...

Thanks for the suggestion. The sentence was reformulated as follows:

'Considering several options (e.g. linear ramp-up), we prescribed a constant land-cover for the first 850 years of the past2k runs.'

9. 'The tree-ring sites are displayed in Fig. A1. For the model-tree-ring comparison a land mask was applied to the model 2m air temperature analyzing the NH extratropics between 40°N and 75°N.'

Here and in other instances in the manuscript, it is not clear whether the reconstructed NH temperature was just calculated as the to be the simple average of the local temperature reconstructions at the tree-ring sites or whether there was a more sophisticated reconstructions method, for instance by calibrating a statistical model to replicate the NH mean temperature ( as in Stoffel et al. 2015). The present manuscript lists in Table 2 just 6 records. Is the NH temperature the average of only these 6 records or is it the temperature reconstructed by Stoffel et al. ? I think that the simple average of these 6 records cannot meaningfully be considered a Northern Hemisphere average.

Thanks for bringing this to our attention. The temperature reconstructions are used directly from Stoffel et al, and therefore have the statistical model included. We changed to using the data from Büntgen et al. (2021), where the post-processed data was publicly available. The statistical methods are therefore as described by Büntgen et al. (2021). This has been clarified in the methods section.

10. 'Towards the end of the simulation period the ensemble shows a larger spread than at the beginning of the simulations, which corresponds to the ocean heat content state being more different between members in the end than at the beginning of the simulations.'

Could this be an indication that the ensemble set-up is not adequate to investigate the main objective of the study, and that the spread of ocean initial conditions is too narrow ?

The ensemble spread being larger at the end of the simulations is mainly apparent in the ocean heat content. For our proposed mechanism the AMOC and the ocean heat transport is more important, which shows a similar range of variability. We cannot rule out that starting from an earlier point in time, or using an ensemble would give a more different ocean state between the ensemble members. However, to carry out an ensemble, we would have needed an ensemble of past2k runs, which was and is not available and not feasible. We have an included a paragraph on this in the new discussion section (see our response to remark #23 below).

11. The results section includes several paragraphs that actually would belong to a (missing) Discussion section. An example is this paragraph:

'Zhong et al. (2011), and Miller et al. (2012) argued that the ocean - sea-ice feedback could play a major role in sustaining a century long cooling after a cluster of four volcanic eruptions in the mid 13th century. In contrast to these studies, we simulate a multi-decadal sea ice response in the mid 6th to 7th century. '

Consider also this reformulation: In contrast to these studies, our simulated cooling is shorter and lasts only for a few decades.

The manuscript was written in a way that the results and discussion were placed in one section. The figures were first described and then discussed in order. We have now split the results and discussion section as suggested.

The sentence given as an example has been reformulated as follows:

'In contrast to these studies, our simulated cooling is shorter and lasts for several decades.'

12. After the 536/540 CE double event, the ensemble mean of the model simulations does not return to zero sea-ice cover anomalies before 560 CE.

the ice-cover in the ensemble mean takes longer to recover and only reaches the climatological mean value by year xxxx.

Thanks for your suggestion, we have taken your comment into account, and now refer to the 0-1850 CE mean.

13. 'Fig 3 (TRW) and triangles (MXD) in the 2 m air temperature maps. The  $2\sigma$  ( $1\sigma$ ) standard deviations for 2m air temperature SLP, evaporation (and precipitation) are stippled (hatched). '

The ensemble standard deviation sigma (2xsigma) ... are stippled (hatched)

The comment (figure caption) has been taken into account, and has been corrected.

14. 'there is a land-sea contrast present for evaporation in summer, where the signal is opposite over the ocean.'

In summer, the sign of the evaporation anomalies over ocean and land is opposite.

The sentence (line 195 in the original manuscript) has been reformulated as follows:

'In summer, the sign of the evaporation anomalies over land and ocean is opposite.'

15. 'the north side of the climatological high pressure systems reflecting an atmospheric circulation separation at around 45°N.'

this sentence is unclear. Does it mean that the sign of the anomalies is apposite north and south of 45N?

Indeed, it means the sign of the anomaly is opposite north and south of 45N. We have separated the results and discussion sections, and we have taken your comment into account when rewriting the sections.

16. 'The long term response is shown on the right side of Figure 3.'

on the right side or in the right half of the picture. Better still is to label all the panels and refer to them accordingly.

The comment has been taken into consideration, and the figure has been changed accordingly. In addition, for better readability of the figure, we have separated Figure 3 into two figures (Fig 3 and 4), see also our reply to reviewer 2.

---

## Author Comment (AC2)

Dear editor and reviewers, thank you for the detailed review of the paper. We have compiled a point by point response going through all comments of the four reviewers. Our comments are given in blue and bold text.

**General reply:**

We have considerably revised the manuscript by rewriting the abstract, introduction and summary and conclusion, as well as separating the discussion from the result section. We updated the references in the introduction and discussion. As suggested, we now use the Büntgen et al. (2021) tree-ring reconstruction ensemble for the model - tree-ring comparison (see figure below) and have rewritten the comparison section accordingly, focusing on the Northern Hemisphere temperature anomaly.

**Reviewer 2:**

This is an interesting paper showing new model based information on seasonal climate response following major volcanic events during the 6th and 7th century. In addition, the paper shows also proxy based reconstructions for different regions of the NH and a few simple model/reconstruction comparisons.

There are a couple of shortcomings, gaps and weak parts of the manuscript that need to be revised, I thus recommend major revisions

Thanks for your constructive comments. We have taken your comments into account in the revised manuscript, as described in the general comments. The detailed comments are answered below.

Comments (in no particular order):

**Abstract**

1. The abstract needs a major revision, currently it is a compilation of individual results without a clear statement. There are statements about proxy reconstructions, model output and model-data

comparison, but there is no connection between the sections and also no dynamic explanations of what was found and new conclusions about climate during the 6/7 century.

The abstract was rewritten accordingly.

2. The first sentence is confusing as there is no actual detection/attribution study (yet) to support the finding that volcanoes are the main cause of cooling and not natural variability or a combination thereof.

We have changed the formulation of the sentence.

'While the onset of this cold period can be clearly connected to the volcanic eruptions in 536 and 540 Common Era (CE), the duration, extent and magnitude of the cold period is uncertain.'

3. Further, several paleoproxies are mentioned, but it is not clear what they refer to, whether they are used to date volcanic eruptions or to derive past climate.

The multiple paleoproxies, ice-core records, tree-rings and historical documents, are referring to both the past climate and the date of the volcanic eruptions. We elaborate on these in the introduction, and would like to leave it in the abstract as it is.

**General manuscript**

4. It would be interesting and valuable to have a paragraph on the socio-economic impact of the cooler and more variable climate during this period. This would be a nice link to the other sections of the paper.

We have a small paragraph about our future work at the end of the conclusion, as we want to investigate this in our next study. See also our reply to reviewer 1.

5. The paper is not up to date with the latest publications and the introduction needs to be revised accordingly. The first paragraph of the introduction is incomplete, confusing and needs better focus and the inclusion of more appropriate and new references to the state of the art in paleo-reconstructions for the study period.

Thank you for your suggestion. Newer publications have been added to the introduction, and the section has been rewritten, taking your first comments into account. The first paragraph has been restructured so that it is better focused and we added new references for the paleoreconstruction part. We now cite the new Pages2k consortium (2017) paper instead of the previous one, and papers from a.o. Moreno Chamarro et al. (2017) and Büntgen et al. (2021) have been added as well.

6. In general, the paper needs to be updated with recent findings related to reconstructions, post-volcanic responses in the paleocontext, regional interpretations, and data/model comparison studies.

Thanks for your suggestion. The part on reconstructions in the introduction has been better explained. The Büntgen et al., (2021) tree-ring reconstruction ensemble has now been used for the proxy comparison and we, therefore, focus the model - tree-ring section mainly on the new NH compilation. The regional tree ring comparisons figures have been moved to the appendix. The papers of Moreno-Chamarro et al. (2017), Zhu et al. (2020), and Halloran et al. (2020) have been added to the discussion to update the section with newer publications.

7. Furthermore, the paper also needs considerations on uncertainties.

Thanks for the suggestion. Now, we address the uncertainties in volcanic forcing, using different models, and the model - tree-ring comparison in the new discussion section.

8. In addition, two studies from 10 years ago on the onset of the LIA are cited (lines 39 ff). They are superseded by new findings.

**The LIA is not in the focus of our study, and we do not aim to make a direct comparison. However, we added one more paper by Moreno-Chamaro et al. (2017) on the LIA period.**

9. It is not clear why Stoffel et al., Büntgen et al. and Esper et al. are used for comparisons with the model (Fig. 5). New reconstructions are available (see Büntgen et al. 2021 Nat. Comm for a review) and possibly an ensemble series could have been used instead of single reconstructions, which do not reflect the true NH conditions but are locally biased.

At the time of writing and submitting our paper, the study by Büntgen et al. (2021) was not published yet. However, we acknowledge the importance of using the newest data available and have therefore reanalyzed the model – tree-ring comparison with the Büntgen et al. (2021) tree ring reconstruction ensemble data. The figures (Fig. 5 in the original manuscript) and corresponding text (Section 3.4 in the original manuscript) have been replaced and rewritten respectively.

10. It is also not clear why the authors compare the grid-based model output with local tree-ringbased reconstructions for three regions. In lines 390ff. they mention that such a comparison can be misleading, so they might reconsider this section. Since we now use Büntgen et al. (2021), the focus in the main manuscript is on the Northern Hemisphere and we no longer focus on the local tree-ring-based reconstructions.

11. A more appropriate comparison could be made with continental reconstructions for Europe (Luterbacher et al. 2016, Env. Res. Lett) and for Asia (Zhang et al. 2018 Nat. Sci. Reports, 8, 7702).

See our response to comment 9 above. We now use the Büntgen et al. (2021) tree-ring reconstruction ensemble. We, therefore, focus on the Northern Hemisphere compilation in the main manuscript.

12. The *methods* and associated measures to compare reconstructions and model output need to be explained in more detail.

Thanks for your suggestion. Since we use the data from Büntgen et al. (2021) instead we rewrote the methods section accordingly. We have taken up your comment and added some details in the method section for the comparison. All the details of the tree-ring reconstruction ensemble can be found in the Büntgen et al. (2021) paper.

13. The authors report on the summer precipitation behaviour in the Mediterranean region in the model world. This part needs to be revised, as in reality there is hardly any precipitation in the warm season and if there is, it is mostly on the northern rim. Even in post-volcanic summers there is no clear signal in observations and reconstructions of the last centuries (Wegmann et al. 2014; Fischer et al. 2007, CRL). Please note that there are hydroclimate reconstructions from different areas of the Mediterranean with which the model output can be compared.

Both Wegmann et al. (2014) and Fischer et al. (2007) describe an increase in precipitation over southern Europe in summer, though a weak one. The modelled increase in precipitation in the Mediterranean is significant and corresponds to a shift in the larger scale atmospheric circulation, which is why we take it up in our discussion. The aim of this paper is to investigate if we simulate a long lasting cooling after volcanic eruptions and its mechanisms in general. Thus, we do not want to go in detail and compare single model variables with limited area-wide hydroclimate reconstructions, which is beyond the scope of this paper.

We have now included the papers from Fisher et al. (2007) and Rao et al. (2017), as suggested by you and reviewer 3.

14. In general, the work has a bias towards summer, which is not surprising as the tree-ring reconstructions resolve summer conditions. However, it would be important to provide some more insight into the conditions during the cold season and how the volcanic influence could change the annual cycle after the short and decadal volcanic influence.

The winter response is also described, especially when we connect it to the ocean - sea-ice feedback during winter/spring. In the discussion, there is more focus on the surface climate response in summer, because the patterns are significant, as opposed to the winter patterns, that are less significant due to the large internal variability in the winter months. To go into more detail of the winter circulation response would be interesting but goes beyond the scope of this paper here.

**Section 3.1. volcanic response:**

15. It seems that the Figures 2,a, c and d are not commented and interpreted in the main text. Are the time series in Figure 2 referring to summer?

Thanks for the comment, the description of figure 2a, c and d have been added to the results. Figure 2 refers to yearly mean responses and this has been elaborated in the text.

16. The plots are small and details cannot be seen. Please could you increase the readability of the figures in general, thank you.

The figure sizes have been increased, and figure 3 has been separated into two figures, so the maps are better readable. The sea-ice extent now is a separate figure.

17. It is not entirely clear what Figure 3 shows. Are all post-volcanic years during the study period averaged and shown in relation to pre-volcanic non-volcanic conditions? Please state this more clearly in the caption and main text. Also, please provide more explanation of the multi-decadal analysis and how it is carried out.

The new figure 3 (shown below) shows indeed an average of all the post-volcanic years, in relation to the 0-1850 CE mean. The 0-1850 CE mean was taken because this is a long enough period for the volcanic response to be negligible. Test calculations were carried out to check the difference between the mean of the 0-1850 CE run and the mean of the 1200 year control run (without volcanic forcing). The difference was very small and not significant. Therefore, the 0-1850 CE mean was used to calculate the anomalies. This has been elaborated in the Results section and the caption. Some sentences have also been added to the methods to clarify the calculations.

---

## Author Comment (AC3)

**Dear editor and reviewers, thank you for the detailed review of the paper. We have compiled a point by point response going through all comments of the four reviews. Our comments are given in blue and bold text.**

**General reply:**
**We have considerably revised the manuscript by rewriting the abstract, introduction and summary and conclusion, as well as separating the discussion from the result section. We updated the references in the introduction and discussion. As suggested, we now use the Büntgen et al. (2021) tree-ring reconstruction ensemble for the model - tree-ring comparison (see figure below) and have rewritten the comparison section accordingly, focusing on the Northern Hemisphere temperature anomaly.**

[Figure]

**Reviewer 3:**

This paper develops and analyzes an ensemble of climate model simulations covering the period of 4 large eruptions in the 5th and 6th century as well as the decades following. These modeling results are really very exciting because they provide brand new and much needed insight into the potential behavior of the climate system following large and important eruptions (including two closely-spaced eruptions in the 6th century) in the first millennium and the potential for new paleoclimate proxy/model comparisons of this important but still sparsely known period of the Common Era. These results will therefore be of great interest in understanding the range of responses to volcanic eruptions and relevant for both modelers and proxy paleoclimatologists.

My primary critical observation is that the manuscript is excessively descriptive. I think a stronger manuscript would result from placing the model (and the proxy-model) comparison directly in the larger context of uncertainties about first millennium climate and response to large eruptions, isolating the model behavior that is most interest to understanding this unique time period, a more complete treatment of the proxy data, and more accounting for uncertainty (the real strength of the model ensemble is the range of possibly behaviors of the ocean-atmosphere system that can be

observed and used to quantify variability in the response). My major comments below are mostly about these issues, and some minor comments and suggestions follow.

**Thank you for your constructive comments. We have taken your comments into account in the revised manuscript, as further answered above in the general comments and in detail below.**

**We have revised the manuscript so that it is less descriptive by separating the result and the discussion parts, and we elaborated on the uncertainties about first millennium climate reconstructions and model behaviour. We now use the Büntgen et al. (2021) tree-ring reconstruction ensemble for the model - tree-ring comparison, which consists of an ensemble of reconstructions, which reduces the uncertainty of individual temperature reconstructions.**

Major comments:

1. Abstract and Introduction: I think the paper (and readers) would benefit from a re-writing of the abstract and a re-framing of the importance of this work.

**Thank you for your suggestion. The abstract was rewritten, re-framing the importance of this work by taking your and reviewer 2' suggestion into account. For example:**

**'While the onset of this cold period can be clearly connected to the volcanic eruptions in 536 and 540 Common Era (CE), the duration, extent and magnitude of the cold period is uncertain.'**

2. As it currently stands, one of the particularly interesting aspects of the 6th/7th century eruptions (that persistence or not of the cooling and therefore the duration of the 'Late Antique Little Ice Age') is mentioned only rather vaguely, and much of the abstract is given over to a simple description of the model phenomena.

**We have reframed the abstract in a way that the motivation of the work is more clear, and we have described the results in a way that they are more interconnected.**

3. A stronger abstract would set the stage for the proxy disagreements (and note the sparseness of the proxy data as exacerbating the difficulty in understanding these uncertainties) and then frame the results in terms of this as well as the importance of understanding large events like the closely spaced eruptions of the 6th century.

**Thanks for your very good suggestion. We have rewritten the abstract accordingly.**

4. Likewise, the Introduction would benefit from using more recent publications about the Little Ice Age as well as a more structured framing of the uncertainty and motivation for the study.

**The paper by Moreno-Chamarro et al. (2017) has been added to the introduction and discussion sections. The LIA is not the focus period of our study, and we do not aim to make a direct comparison, as we do not have this period in our model simulations. We therefore have added only one newer publication on the LIA period, as we do discuss the similar mechanism behind the cooling and thus agree it is important for the discussion part.**

**As described in our overall reply, we have revised the manuscript extensively, with new tree-ring data comparison, a rewritten abstract, introduction and summary and conclusions, and elaborated discussion sections.**

5. Initial discussion of proxy data (Lines 53 to 59). This section needs to be enhanced, as an appreciation for the uncertainties and causes of uncertainties for first millennium climate reconstructions, particularly with the resolution needed to resolve volcanic eruptions precisely, is important for the comparisons that come later in the paper.

**The paper of the Pages2k consortium (2017) has been added to address the uncertainties for first millennium climate reconstructions.**

6. While Ahmed et al. 2013 (which is properly PAGES 2k Network 2013) was a distillation of temperature proxy data for the last IPCC, it is superseded by a number of papers, including PAGES2k Consortium 2017 (Emile-Geay et al. 2017).

**We updated the section with the paper of Pages2k consortium (2017).**

7. The authors might also want to cite Esper et al. 2018 (doi is 10.1016/j.dendro.2018.06.001) which analyzes tree-ring proxy uncertainties in the early part of the last millennium (and therefore these uncertainties will be even greater in the first millennium of the Common Era). Particularly here: there is a substantial body of literature now (some of it discussed later) about the ability of different tree-ring proxy measurements to resolve or 'smear' volcanic cooling - MXD vs tree-ring width.

**Thanks for your suggestion. The study has been included and is now used to address the tree-ring uncertainties and sparsity in proxy data in the first millennium in the discussion section.**

8. Similarly, multiproxy approaches that mix seasons, hemispheres, or are low resolution might not resolve the volcanic signal or may require additional post-processing of model data to make an appropriate comparison. Since the source of possible uncertainties in proxy reconstruction of

6th/7th century climate is important for the comparisons that come later, I think a more thorough and up-to-date discussion is warranted here in the introduction.

**Thanks for your suggestion. We have added why a multiproxy approach would not be the best to use for this study in the introduction.**

9. Proxy data (Section 2.2): As above, I think a more complete and clear description of the proxy data here would be useful for later in the paper when comparisons become important.

**Since we use the data from Büntgen et al. (2021) instead we rewrote the methods section accordingly, also taking up your comment. We point to the corresponding papers for the details of the tree-ring data.**

10. Table 2 lists the individual proxy data that are available (this is good to have this, since the representation of tree-ring width and MXD can sometimes be subsumed when using a reconstruction), but the wording in Section 2.2 is confusing - for instance, what does 'The first four sites combined are the "NH land" compilation by Stoffel et al. (2015)' mean? Does this mean that the 4 sites listed first in the Table were also used by Stoffel? This isn't clear.

**Thanks for bringing this to our attention. The first four sites were indeed used by Stoffel et al. (2015). Since we changed the tree-ring data to Büntgen et al. (2021), the table is updated accordingly and moved to the appendix.**

11. By the time one arrives at Figure 5 and the associated text, it isn't clear what/which of each of these proxies is going into the comparison, so a more thorough discussion of the proxy data used and what each reconstruction in Figure 5 contains is necessary.

**The tree-ring part in the results and discussion section has been rewritten according to the new ensemble reconstructions.**

12. The following line says 'The data sets 135 contain a mix of tree ring width (TRW) and maximum latewood density (MXD).' and this is true, but only the NSCAN MXD data are available for the 6th and 7th century - the rest are tree-ring width and subject to the potential problems described in the following lines. Again, this section seems rather sparse and is not clearly organized, and yet limitation of the proxy data (or their particular time series properties) will become important later in the paper. Since the authors prepared this manuscript, there as been a new ensemble reconstruction of Common Era temperature (Buntgen et al. 2021, doi is 10.1038/s41467-021-23627-6) - while I realize this paper actually came out after this manuscript was submitted and the authors cannot have been expected to use these reconstructions (of course!) I would encourage them

to at least consider them (formally analytically or informally as comparison, since the early part of the LALIA is examined in the paper), at the discretion of the editor.

**We agree with this comment. Thus, the new data set from Büntgen et al. (2021) has been used in the revised manuscript, and the tree-ring section has been rewritten, as mentioned above.**

13. Finally, it would be worth I think mentioning why multiproxy reconstruction like PAGES2k, LMR etc. are likely to be unsuitable for this comparison and why the authors rely (and rightly I think) on tree-ring data.

**Thank you for your suggestion. Why a multiproxy reconstruction like PAGES2k etc. is not suitable for this study has been added to the introduction.**

14. Proxy-model comparison: This section is unclear in places and speculative without support in others; for instance, (Line 355) I'm not sure what 'The temperature anomalies from the model simulations and the 2 sigma variability range fall within the 2 sigma variability of the NH of the model simulations' means? I also find it to be too qualitative - what does 'good agreement' mean and how to measure it?

**Thanks, this was a writing error, it has been corrected.**

**As for the qualitative 'good agreement', we compare the range of variability of the model to the range of variability of the tree-ring ensemble.**

**'... the reconstructed NH temperatures from the ensemble mean fall within the spread of the model simulations.'**

**The proxy-model comparison section was rewritten, as we now use the tree-ring reconstruction ensemble from Büntgen et al. (2021). We have taken your comment into account in the comparison, by using absolute numbers and the standard deviation.**

15. In Line 362, 'More deviation is visible' is also vague. I think this section would benefit from a more straightforward and quantitative exploration of the proxy-model comparisons.

**The comparison section has been rewritten according to the new Büntgen et al. (2021) tree-ring reconstruction ensemble we are now using. We have taken your comment into account in the revision, see our previous comment.**

16. In Line 365, I'm not sure how something could be both 'less good ... but still remarkable'? However, also the full range of the model ensemble should really be considered in the comparison -

the 'real world' is just simply one iteration of what could have happened under different initial conditions, forcing uncertainty, feedbacks, and interactions and stochastic variability. So the comparison is not simply to the multimodel ensemble mean or even peak cooling, but taking into account the full range of ensemble variability and seeing the tree-ring data as one 'ensemble member' of possible actual and physically plausible atmosphere-ocean states.

**Thank you for your suggestion. As for the above comment, we have made the comparison discussion more quantitative, by describing the comparison in terms of significance and anomaly values. In addition, the ensemble range has been taken into account in the rewritten comparison discussion.**

17. Later in Line 376, the authors write that 'There is a good agreement between the tree-ring temperatures and the model temperatures after normalization' - but again this lack of precision doesn't do justice to the comparison - indeed there appears to be reasonable association for the major eruptions for NH temperature from Stoffel (including some MXD) and the NSCAN MXD, but for Alps and Altai the lag recovery is longer. So simply saying there is a good agreement masks interesting differences.

**We have taken your comment into account and have elaborated the discussion of the differences between the model and the tree-ring ensemble. Since we are now using the Büntgen et al. (2021) tree-ring reconstruction ensemble and to be able to go more in detail when it comes to agreements and discrepancies, we decided to focus on the NH in the main manuscript and to move the comparison on the individual sites to the appendix.**

18. In Line 381, this seems very highly speculative: 'could be due to the prescribed volcanic forcing in the model, and that the 547 eruption might have had a stronger impact on NH land than the model simulates.' - why wouldn't the same apply to Stoffel or NSCAN then? There would need to be some support for this to claim it as a source of the discrepancy.

**Since we now use the Büntgen et al. (2021) tree-ring reconstruction ensemble, the focus in the main manuscript is on the Northern Hemisphere. The local tree-ring based reconstructions figures have been shifted to the appendix and the corresponding text has been rewritten accordingly. There is a lag after the 536/540 CE eruptions visible in all tree-ring reconstructions, in Nscan as well, though not as pronounced. This could be due to the MXD data that was used for this site. We have taken up your comment and elaborated the explanation of the 547 CE eruption in the discussion.**

19. On Line 390, again this seems highly speculative: 'Perhaps the century long lasting cooling may be only apparent in the Alps and Altai tree-ring records, as the cooling is a local feature occurring at high altitude of the mid-latitudes.'

**There are other signs that the cooling lasted longer in the Alps, for example an advance in glacier fronts. The LIA was spatially heterogeneous in duration and timing and so the same can be true for the LALIA. Records from Greenland ice-cores for the study period (Sigl et al. 2015) agree with the tree-ring records from the Alps (Büntgen et al. 2017). Interestingly, our model simulations reveal a spatial pattern with a hemispheric dipole dividing the NH response at around 45 N. Further tree ring based studies on spatial patterns are needed in the future.**

**We have clarified this point accordingly in the revised manuscript.**

20. Again, in Line 395 the authors speculate that 'The change in hydro-climate corresponds to the soil moisture availability for the trees and thus could have impacted tree ring growth', but again this is just speculation, and indeed for the Alps, which have the longest lag at odds with the model, the 20 years summer precipitation anomaly (Figure 3c) is positive and the winter signal is mixed.

**We have added the papers of Basset et al. (1964) and Müller et al. (2016) to support the theory that moisture availability could have impacted the tree-ring growth. The Alps do indeed have a slight increase in precipitation for the 20 year mean, but they also have an increase in evaporation at the same time. We find the atmospheric circulation separation interesting, and think it is worth further investigation.**

21. Potentially the most parsimonious answer is that tree-ring width has a tendency to increase the 'tail' of the post-volcanic cooling and change the timing of recovery to baseline. But the authors give this only a brief mention in Line 371.

**Thank you for your suggestion, we have elaborated on this in the discussion of the model tree-ring comparison.**

22. Summary and Conclusion: This section is largely a restatement of the paper, but would be stronger with a distillation of the main points of the article and major conclusions.

**The Summary and conclusions have been shortened and rewritten.**

Additional Comments:

23. Line 12, Line 13 'land–sea'

**This has been corrected.**

24. Line 50: perhaps 'multidecadal cooling, as has been reconstructed by Buntgen et al. (2016).'

**The sentence has been altered according to the comment:**

**'...multidecadal cooling, as has been reconstructed (Büntgen et al., 2016; Helama et al., 2017)...'**

25. Line 55: 'Common Era'

**'Common Era' has been introduced in the abstract, after which it is addressed as 'CE'.**

26. Line 101-102: this sentence seems out of place? 'A common paleo-model set-up is to use 1850 pre-industrial conditions, due to model simulation limitations'.
**We have changed the sentence to the following:**

**'In contrast to other studies, where the initial state was simply taken from a pre-industrial control simulation, our approach allows us to include the climate and forcing history of the previous decades and centuries as well as their integrative effects (e.g., Gleckler et al., 2006).'**

27. Line 103: This is confusing as written - but why use standard deviations instead of the temperature deviations in the bootstrap?

**We are not sure what is meant exactly by this comment. We calculated the standard deviations for the bootstrap for each of the variables, and used this to calculate the significance.**

28. Figure 1: are these data from Jungclaus et al. and Toohey and Sigl? Best to include a citation with the figure caption as well.

**The figure caption has been revised:**

**'...Global zonal mean volcanic forcing (Aerosol Optical Depth, AOD) for the study period (520-680 CE) and b) zonal mean accumulated AOD (520-680 CE) in 15 year bins from the reconstructed volcanic forcing of Toohey and Sigl (2017)...'**

29. Table 1: I assume the months (January) are assigned because the actual month of the eruption is not known for these? It would be worth mentioning this (and some of the potential consequences, e.g. Stevenson et al. 2017) in the methods section as well
**Thank you for the suggestion, we have taken it up and added it to the method section.**

**'... The eruptions are set to January in the model forcing as the actual eruption month is unknown.'**

**As for the discussion on eruption season, we have added the Toohey et al. (2011) paper.**

30. Line 152: '2 K' - relative to which baseline? I presume the 521-680 CE mean mentioned in Line 142, but please clarify

**The 2K cooling is relative to the 0-1850 CE mean, which is taken as the baseline throughout the paper. This has now been clarified in the text as follows:**

**'...with a peak cooling of the NH 2m air temperature of ∼2 K … compared to the 0-1850 CE mean.'**

31. Line 157: 'decreased for ∼20 years' - it is difficult to see this in Figure 2 because of the scale - can you provide a range of the actual return to baseline periods? Particularly since the closely-spaced 536/540 eruptions would be expected (I think) to collectively show a longer recovery time than the individual eruptions in the 570s and 626 event

**The actual return periods have been added to the text:**

**'...reveals a maximum cooling in the first and second year after the eruption and is decreased for 20 years after the 540 CE, for 30 years after the 574 CE and for 14 years after 626 CE eruptions, much longer than the direct response of the AOD.'**

32. Line 169-175: Some other more recent papers to consult (and cite as appropriate) might include Lehner et al. 2013 and Slawinska and Robock 2018

**These papers have been taken into consideration and we added the paper of Lehner et al. (2013) to the discussion of ocean - sea-ice feedbacks.**

33. Line 270: Perhaps useful to consult Fischer et al. 2007 (doi is 10.1029/2006GL027992, 2007) and Rao et al. 2017 (doi is 10.1002/2017GL073057) with respect to Mediterranean (and Europe) response to eruptions in historical and paleoclimate data.

**Thanks for the suggestion, we have consulted these papers with respect to the Mediterranean response and included them in our discussion.**

34. Figure 5 caption: 'Fennoscandia'?

**The term Fennoscandia has been replaced by 'Northern Scandinavia (Nscan)'.**

35. Line 372: as well as estimate of the forcing, spatial distribution of AOD anomalies, feedbacks, uncertainty in timing of the eruption (Stevenson et al.) - so, lots of potential uncertainties.

**There are several sources of uncertainties, which comes with using models and proxy data. Uncertainties are described in the new discussion section of the manuscript.**

36. Line 377-: I'm not sure this requires any further explanation - the models and data have different references periods and likely different means, but what is of interest is the volcanic signal, so a renormalization isn't that remarkable

**Thanks for your suggestion, we have taken up your comment. We have now used 0-1850 CE as a reference period for all data series, so that it is consistent throughout the manuscript. In the new discussion we do not go in detail on this realigning of the data.**

**References**

Bassett, J.R., 1964. Tree growth as affected by soil moisture availability. *Soil Science Society of America Journal, 28*(3), pp.436-438.

Büntgen, U., Myglan, V.S., Ljungqvist, F.C., McCormick, M., Di Cosmo, N., Sigl, M., Jungclaus, J., Wagner, S., Krusic, P.J., Esper, J. and Kaplan, J.O., 2016. Cooling and societal change during the Late Antique Little Ice Age from 536 to around 660 AD. *Nature geoscience, 9*(3), pp.231-236.

Büntgen, U., Myglan, V.S., Ljungqvist, F.C., McCormick, M., Di Cosmo, N., Sigl, M., Jungclaus, J., Wagner, S., Krusic, P.J., Esper, J. and Kaplan, J.O., 2017. Reply to 'Limited Late Antique cooling'. *Nature Geoscience, 10*(4), pp.243-243.

Büntgen, U., Allen, K., Anchukaitis, K.J., Arseneault, D., Boucher, É., Bräuning, A., Chatterjee, S., Cherubini, P., Churakova, O.V., Corona, C. and Gennaretti, F., 2021. The influence of decision-making in tree ring-based climate reconstructions. *Nature communications, 12*(1), pp.1-10.

Gleckler, P.J., AchutaRao, K., Gregory, J.M., Santer, B.D., Taylor, K.E. and Wigley, T.M.L., 2006. Krakatoa lives: The effect of volcanic eruptions on ocean heat content and thermal expansion. *Geophysical Research Letters, 33*(17).

Lehner, F., Born, A., Raible, C.C. and Stocker, T.F., 2013. Amplified inception of European Little Ice Age by sea ice–ocean–atmosphere feedbacks. *Journal of climate, 26*(19), pp.7586-7602.

Moreno-Chamarro, E., Zanchettin, D., Lohmann, K., Luterbacher, J. and Jungclaus, J.H., 2017. Winter amplification of the European Little Ice Age cooling by the subpolar gyre. *Scientific Reports, 7*(1), pp.1-8.

Müller, M., Schwab, N., Schickhoff, U., Böhner, J. and Scholten, T., 2016. Soil temperature and soil moisture patterns in a Himalayan alpine treeline ecotone. *Arctic, Antarctic, and Alpine Research, 48*(3), pp.501-521.

PAGES2k Consortium, 2017. A global multiproxy database for temperature reconstructions of the Common Era. *Scientific data, 4.*

Sigl, M., Winstrup, M., McConnell, J.R., Welten, K.C., Plunkett, G., Ludlow, F., Büntgen, U., Caffee, M., Chellman, N., Dahl-Jensen, D. and Fischer, H., 2015. Timing and climate forcing of volcanic eruptions for the past 2,500 years. *Nature, 523*(7562), pp.543-549.

Stoffel, M., Khodri, M., Corona, C., Guillet, S., Poulain, V., Bekki, S., Guiot, J., Luckman, B.H., Oppenheimer, C., Lebas, N. and Beniston, M., 2015. Estimates of volcanic-induced cooling in the Northern Hemisphere over the past 1,500 years. *Nature Geoscience, 8*(10), pp.784-788.

Toohey, M., Krüger, K., Niemeier, U. and Timmreck, C., 2011. The influence of eruption season on the global aerosol evolution and radiative impact of tropical volcanic eruptions. *Atmospheric Chemistry and Physics, 11*(23), pp.12351-12367.

Toohey, M. and Sigl, M., 2017. Volcanic stratospheric sulfur injections and aerosol optical depth from 500 BCE to 1900 CE. *Earth System Science Data, 9*(2), pp.809-831.

---

## Author Comment (AC4)

**Dear editor and reviewers, thank you for the detailed review of the paper. We have compiled a point by point response going through all comments of the four reviews. Our comments are given in blue and bold text.**

**General reply:**
**We have considerably revised the manuscript by rewriting the abstract, introduction and summary and conclusion, as well as separating the discussion from the result section. We updated the references in the introduction and discussion. As suggested, we now use the Büntgen et al. (2021) tree-ring reconstruction ensemble for the model - tree-ring comparison (see figure below) and have rewritten the comparison section accordingly, focusing on the Northern Hemisphere temperature anomaly.**

[Figure]

**Reviewer 4:**

My review of submission cp-2021-49 by van Dijk and co-authors focuses on the paleoclimate part of the paper and not on the models, as the latter are not my field of expertise. In their contribution, the authors seek to answer the question whether long-lasting cooling occurred over the Northern Hemisphere following a cluster of large 6[th] and 7[th] century eruptions which occurred in 536, 540, 574 and 626 CE according to sulfur deposits in bipolar ice cores. The authors do so by comparing proxies (mostly tree-ring reconstructions) with model output.

**Thank you for your constructive comments. We have taken your comments into account in the revised manuscript, as further answered above in the general comments and in detail below.**

The idea of the paper is nice, but I have a few major comments that shall be addressed by the authors in a revised version:

1. The title is misleading, no proxy record has hitherto posited that the 6[th] century eruptions would have been at the origin of a long-lasting hemispheric cooling. Instead, a study based on data from the Alps and Altai (by Buntgen and colleagues, 2016) has just pointed to marked cooling at these two sites. Other tree-ring records do not suggest a comparable cooling. Speaking of hemispheric cooling is thus an overstatement and should be changed.

**We have discussed this comment thoroughly and would like to stay with the given title.**

**The main scope of the paper is to study the volcanic response with this model set-up and volcanic forcing, not on the tree-rings. For us, the title underlines the aim of our study. In the introduction, the long lasting cooling from tree-ring records and other proxies that have provided evidence for volcanic eruptions in the same period as the long lasting cooling from the tree-rings, act as a motivation. Together they lead to our question: Whether a series of volcanic eruptions induced a multidecadal to centennial cooling in the mid 6th to 7th century. The point that the Alps and Altai are not representative for the entire Northern Hemisphere is fair and we have thus altered the aim of the paper accordingly.**

2. Along the same line, starting from line 32ff the authors state that cooling might have exceeded that of the LIA and focus on two site chronologies that were presented in 2016. While the authors rightly present the results of this study, and add the reply provided by Helama and colleagues from 2017, they ignore a vast body of proxy studies that have been published on the topic and where the chronologies cover many sites of the NH. By focusing only on the LALIA study, they ignore a large body of spatial and temporal reconstructions covering the period of interest. The authors should therefore present a more balanced assessment of the existing data by including e.g., Schneider et al. (2015, ERL), NTREND (spatial and temporal; Wilson et al., 2016 QSR ; Anchukaitis et al., 2017 QSR), Guillet et al. (2017 NGEO) or the most recent TRW-based paper from the tree-ring community published lately by Buntgen et al (2021) in NCOMM.

**Thanks for your very good suggestion, which has motivated us to switch using the most recent TRW-based ensemble reconstruction from the tree-ring community (Büntgen et al, 2021). The introduction has been rewritten and updated with newer publications where necessary.**

3. Chapter 2.2: It is not clear to the reader how the authors did the tree-ring analysis. They provide a long discussion on advantages of MXD over TRW data, but it is very unclear how the authors did the reconstructions and what they did with the data. How were the sites/data chosen? More details need to be provided here as it remains very unclear to the referee how the proxy series were developed.

**Since the new tree-ring reconstruction ensemble from Büntgen et al. (2021) is used in the revised manuscript, we rewrote the methods for the tree-ring part. The data from Büntgen et al. (2021) is publicly available and we used them as published. We have clarified this in the methods section.**

4. The same holds true for the NH approach: why did they not use the spatial reconstruction data from Anchukaitis et al. (2017) or Guillet et al. (2017)?

**See the general explanation at the beginning of the comment replies. We now use the most recent tree-ring reconstruction ensemble from Büntgen et al. (2021).**

5. Another major drawback is the restriction of the comparison of model with tree-ring data (lines 349ff) just between the Alps, Altai (both known for excessive cooling in tree-ring records) and Scandinavia. Why did the authors not rely on the full set of tree-ring reconstructions and include a comparison for Siberia, Central Asia and North America?

**We have done a comparison for the sites for Siberia, Central Asia and North America as well, but both the model and tree-ring reconstructions showed such high variability that the volcanic signals were lost. We therefore chose only to show the sites for the Alps, Altai and Scandinavia. Since we are now using different tree-ring data, we decided to focus on the NH compilation and go more in detail here. The individual sites are added to the appendix and we added this information.**

For the paper to become acceptable, the breadth of the proxy records needs to widened and the methods need to be described in much more detail.

**The main scope of the paper is about the model simulations and the mechanism for a multidecadal volcanic induced cooling. New, published tree-ring data are used now and we described this data in the methods section, referring to the corresponding papers for the details. We have clarified these points in the revised manuscript.**

Minor points:

6. Line 2-4 (Abstract): to which "multiple paleo proxies" are the authors referring to? This is a misleading statement as the proxy records pointing to massive and long lasting cooling are few. This needs remediation

**The multiple paleoproxies, ice-core records, tree-rings and historical documents, are referring to both reconstructing the past climate and the date of the volcanic eruptions. We elaborate on**

**these in the introduction, and would like to leave it in the abstract as it is. The definition of a 'long lasting cooling' includes the decadal scale, which is what several of the proxies are showing as well. Thus, we would like to leave it in.**

7. Line 12/13: "see" should be changed to "sea"

**Thanks, corrected.**

8. Lines 19-21: This needs some rephrasing, stating that the cooling was 20 years in the proxy records is somewhat an overstatement. The initial cooling was in fact there, but temperature recovered rather quickly to more normal conditions and reached "fully normal" after two decades. Some clarification would be good here.

**Thanks, we have taken your comment into account when rewriting the abstract.**

9. Line 30: what lines of evidence do you have to compare the 6th century cooling to the conditions that led to the LIA? I suggest that either references are provided or that this statement is removed.

**Thanks for your suggestion. The introduction has been rewritten, and we have taken the reference to the LIA out of the sentence.**

10. Line 152: How does peak cooling in models compare with proxies? How does the amplitude of cooling compare between the two data sets?

**In this section, we show the results of the model and its indication. The comparison with the proxy data is made in a separate part of the section.**

11. Line 155: what do you mean with background level for AOD? <0.1?

**By background level we mean the 0 line in the AOD plot in figure 2. We have clarified this in the text.**

**'The AOD peaks after ~12 months and is back at 0 after 3-4 years (Fig. 2b).'**

12. Line 376: the LALIA concept is based on records from two sites. The authors should go beyond these sites and analyze all data that exists in the NH. It is unclear why the study is limited to Alps, Altai and Fennoscandia

**We chose these sites to touch upon the contrast between individual sites and a compilation for the entire NH. The other individual sites were hard to compare with, as the variability in both**

**data sets overwhelms the volcanic signal. We have now used the new Büntgen et al. (2021) TRW-based ensemble reconstruction data, and therefore focus on the NH in the main manuscript. The individual sites have been added to the supplementary material.**

13. Line 386: use the data presented in the Buntgen et al. (2021) ensemble study instead

**Thanks for the very good suggestion, we now used the new tree-ring reconstruction ensemble from Büntgen et al. (2021) instead.**

**References**

**Büntgen, U., Allen, K., Anchukaitis, K.J., Arseneault, D., Boucher, É., Bräuning, A., Chatterjee, S., Cherubini, P., Churakova, O.V., Corona, C. and Gennaretti, F., 2021. The influence of decision-making in tree ring-based climate reconstructions. *Nature communications, 12*(1), pp.1-10.**

---

## Author Response (AR1)

**Comments to the author**:
Dear colleagues, thank you very much for addressing the points by the 4 reviewer. Can I please ask you to revise the paper accordingly. Just a few points related to your answers to the reviewers:

**Dear Editor Jürg Luterbacher,**

**Thank you for your comments.**

**We have carefully checked the suggested proxy data sets and have chosen to include a comparison with the PAGES 2k (2019) and the Luterbacher et al. (2016) data sets. We have added those figures to the appendix. Our response to the comments is given in further detail below, in blue and bold text.**

1. On the use of the Buentgen al paper which is certainly the latest published large scale reconstruction and suggested by different reviewers. Another option would have been Neukom et al. that was presented in the latest IPCC AR6. Using and showing only reconstruction is critical as it will not allow a complete and unbiased picture. Also Buentgen et al. has limitations/challenges, is annual with a summer and tree ring only biased I would like to see a more broader comparison including other hemispheric reconstructions (as suggesting by reviewer 4, point 2)

*R4 C2. Along the same line, starting from line 32ff the authors state that cooling might have exceeded that of the LIA and focus on two site chronologies that were presented in 2016. While the authors rightly present the results of this study, and add the reply provided by Helama and colleagues from 2017, they ignore a vast body of proxy studies that have been published on the topic and where the chronologies cover many sites of the NH. By focusing only on the LALIA study, they ignore a large body of spatial and temporal reconstructions covering the period of interest. The authors should therefore present a more balanced assessment of the existing data by including e.g., Schneider et al. (2015, ERL), NTREND (spatial and temporal; Wilson et al., 2016 QSR; Anchukaitis et al., 2017 QSR), Guillet et al. (2017 NGEO) or the most recent TRW-based paper from the tree-ring community published lately by Buntgen et al (2021) in NCOMM.*

**As stated in our previous reply we compare our simulations now with the ensemble reconstruction from Buentgen et al. (2021), which encompasses 15 different NH tree ring reconstructions.**

**The explicit data sets suggested by Reviewer 4 do not cover our period of interest: Schneider et al. (2015) covers 600-2000 CE, NTREND (Wilson et al., 2016; Anchukaitis et al., 2017) covers 918-2004 CE and Guillet et al. (2017) covers the 1257 Samalas eruption. On top, the three latter reconstruction methods are included in the Büntgen et al. (2021) study.**

**According to your suggestions, we have extended our comparison to two different reconstructions. We compare our model results now with the hemispheric scale proxy reconstructions from the PAGES2k consortium (Neukom et al. 2019) for the NH annual mean, as well as for Europe JJA with the reconstruction by Luterbacher et al. (2016), see our answers below. For both reconstructions, we used the composite plus scaling (CPS)**

**method data set. We have added these two figures to the appendix as Fig. A3 and A2 d) and elaborated on them in the discussion and appendix.**

[Figure]

New **Figure A3: Model - proxy comparison for the entire NH annual mean. The proxy data is the PAGES2k consortium (Neukom 2019). Anomalies are calculated wrt 0-1850 CE.**

2. as well as a comparison at continental scale as suggested by reviewer 2, point 11 at least for some of the parts of the manuscript.

*R2C11. A more appropriate comparison could be made with continental reconstructions for Europe (Luterbacher et al. 2016, Env. Res. Lett) and for Asia (Zhang et al. 2018 Nat. Sci. Reports, 8, 7702).*

**When it comes to continental reconstructions, we have now added Luterbacher et al. (2016) to figure A2 in the Appendix (see figure A2 d) below). The data used by Zhang et al. (2018) consist of decadal-mean data and thus does not include the annual peak cooling.**

[Figure]

**New Figure A2** Model - tree-ring comparison for a) Northern Scandinavia (N-Scan) (Esper et al., 2012b), b) the Alps and c) the Altai (Büntgen et al., 2016) and d) Europe (Luterbacher et al., 2016). Near-surface air temperatures from the model are taken for the corresponding area and for JJA and land only. Anomalies are calculated wrt 0-1850 CE.

On the Buentgen/model comparison.: It seems that there is a large difference around 540, around 575 and 630, also obvious is the general overestimation from around 545 to 575 and underestimation until 640. Indeed, it would be interesting if this is a feature for this reconstruction only or also shared with independent evidence from other reconstructions.

**In figure A3, we compare the entire NH annual mean from the model simulations with the PAGES 2k data. The PAGES 2k (2019) data show different signals and responses following large eruptions compared to the model simulations. The temperature response following the 536/540 CE and 626 CE eruptions are weaker in the proxy data. The recovery time after 540 CE however, agrees well with the model simulations. Whereas Büntgen et al. (2021) shows a weaker signal to the 574 CE eruption, the PAGES 2k data reveal no response at all. In addition, the PAGES 2k proxy data is up to 0.5 degrees warmer between 580 and 610 CE.**

**Wrt to the new Figs. A2 and A3, we have added the following text to the ms:**

**4 Discussion**

**Model - tree ring comparison**

…

"Reconstruction records are becoming more uncertain further back in time due to the sparseness of available proxy records especially before 1200 CE (Masson-Delmotte, 2013; Esper et al., 2018; PAGES 2k, 2019). Comparing the model results with temperature reconstructions for the entire NH annual mean (PAGES 2k, 2019) reveals a less good agreement (see Appendix A, Figure A3)."

…

"Previous proxy-based studies (Larsen et al., 2008; Esper et al., 2012b; Luterbacher et al., 2016; Helama et al., 2017; PAGES 2k, 2019) found a cooling up to 570 CE for Scandinavia, Europe, and the NH, based on tree-ring, ice-core, lake sediment, and documentary records (Figures A2a and d, and A3)."

**Appendix A - Model - tree ring, and multi-proxy reconstruction comparison:**

"Most temperature reconstruction data sets go back to about 1200 CE, and the further back in time, the fewer proxy records remain, and the more uncertainties they contain (Masson-Delmotte, 2013; Esper et al., 2018; Neukom et al., 2019). The main proxy type that remains to reconstruct the temperatures in the Northern Hemisphere (and especially mid-high latitudes, Europe) are tree rings (Neukom et al., 2019), and they are often used to reconstruct the temperature in especially Europe (Luterbacher et al., 2016). Other reconstructions available consist of a mix of proxies with a more limited dating precision, which leads to a reduction of the amplitude of the signals (Sigl et al., 2015; Büntgen et al., 2020; Plunkett et al., 2021). Only 25% of the proxies available for our study period have annual dating precision (Sigl et al., 2015).

The data sets used for the individual tree-ring sites are from Northern Scandinavia (N-Scan) (Esper et al., 2012b, a), from the Alps and Altai (Büntgen et al., 2016) and from Europe (Luterbacher et al., 2016) (Table 2)."

**Table 2.** Overview of tree-ring/proxy locations and type (MXD: maximum latewood density; TRW: tree ring width) used in this individual comparison. For the proxy reconstructions from Luterbacher et al. (2016) and PAGES 2k the method used is composite plus scaling (CPS).

| Location name | Coordinates | Type of data | Reference |
|---|---|---|---|
| N-Scan | 66°- 70°N, 19°- 29°E | MXD | Esper et al. (2012b, a) |
| Alps | 46°N, 12.5°E | TRW | Büntgen et al. (2016) |
| Altai | 50°N, 87.5°E | TRW | Büntgen et al. (2016) |
| Europe | 35°- 70°N, -25°- 40°E | TRW and MXD | Luterbacher et al. (2016) |
| NH | 0°- 90°N, -180°- 180°E | mixed proxies | (PAGES 2k, 2019) |

...

"For Europe, the model and proxy data (Luterbacher et al., 2016) agree well. The proxy reconstruction falls within the spread of the model ensemble (Fig. A2d). As for the comparison with Büntgen et al. (2021), the peak cooling is less for the reconstructions, and there is an lag in the proxy data after 540 CE.

To illustrate the reduction in amplitude of the signals, we compared the PAGES 2k multi-proxy reconstruction (Neukom et al., 2019) to the simulated NH annual mean near surface temperature (Fig. A3). The proxy data agrees well for the recovery period after the 536/540 CE double eruption, and from ~650 CE - 675 CE, but the reconstructions show a weaker cooling following the volcanic eruptions, and for the 574 CE eruption the reconstruction does not show a signal at all."

Review 2:

point 11: please see comment above

point 17: I don't agree on the following answer: The 0-1850 CE mean was taken because this is a long enough period for the volcanic response to be negligible.

Within this period there are clearly times with strong volcanic activity. I suggest to show the difference with respect to a shorter period without any large volcanic events.

**In the figure below, we have calculated the temperature anomalies with respect to a) the control run for 0 CE conditions, b) the 0-1850 CE past2k mean, as we have used in our anomaly calculations in the paper, and c) the 525-535 CE mean (a period before without volcanic eruptions) from the past2k run. As you can see, this gives the same result (within +/- 0.15 K) as when using the entire 0-1850 period for the anomaly calculation from the past2k run. We therefore argue that using the 0-1850 CE mean does not affect the results of our study. We have added this information to the Methods section and we have revised the sentence to: "*We use the average over 0-1850 CE to have a reference climate that is representative for the entire pre-industrial Common Era.*"**

[Figure]

**Figure. Ensemble temperature anomalies wrt a) CTR (0 CE), b) 0-1850 CE, and c) 520-535 CE mean of the study period.**

point 18, NAO comparison. This indeed is not clear to me why the model shows the opposite of the reconstructions and what we know from recent strong volcanoes with strong positive NAO during winter within the first few years.

**A positive NAO and a surface winter warming pattern has been observed after large tropical eruptions in the past (Robock and Mao, 1992; Robock, 2000 and following work). However, not all IPCC models show this signal (Stenchikov et al., 2002; Driscoll et al., 2012 and following papers). The cause for this is still highly debated, f.e., discussing model deficiencies (low top versus high top models; Charlton-Perez et al., 2013), volcanic aerosol forcing details (Toohey et al., 2014), strength of the volcanic eruption and forcing (Bittner et al., 2016a), tropical vs high latitude eruption impacts (Schneider et al., 2009), high internal variability and number of model ensemble members (Bittner et al., 2016b), the role of the ENSO state during the eruption (Coupe and Robock, 2021), up to if the observed signal is due to volcanic eruptions at all (Polvani et al., 2019).**

Here, we show that there is a positive NAO in the first winter after the eruptions visible after three out of the four large eruptions during 520-680 CE (old Figure A3). The maps in Figure 3 are a mean of two winters as well as four eruptions times 12 ensemble members, and so the signal is smoothed. We have added these details to the discussion.

point 19: this might be misleading to mention and discuss the non-significant areas. I propose to concentrate on the stat. sign. Differences.

In some cases, like for the NAO and the atmospheric circulation pattern, we would like to show and stress these results due to the ongoing debate in this research field.

References:

Anchukaitis, K.J., Wilson, R., Briffa, K.R., Büntgen, U., Cook, E.R., D'Arrigo, R., Davi, N., Esper, J., Frank, D., Gunnarson, B.E. and Hegerl, G., 2017. Last millennium Northern Hemisphere summer temperatures from tree rings: Part II, spatially resolved reconstructions. *Quaternary Science Reviews*, *163*, pp.1-22.

Bittner, M., Timmreck, C., Schmidt, H., Toohey, M. and Krüger, K., 2016a. The impact of wave‑mean flow interaction on the Northern Hemisphere polar vortex after tropical volcanic eruptions. *Journal of Geophysical Research: Atmospheres*, *121*(10), pp.5281-5297.

Bittner, M., Schmidt, H., Timmreck, C. and Sienz, F., 2016b. Using a large ensemble of simulations to assess the Northern Hemisphere stratospheric dynamical response to tropical volcanic eruptions and its uncertainty. *Geophysical Research Letters*, *43*(17), pp.9324-9332.

Büntgen, U., Allen, K., Anchukaitis, K.J., Arseneault, D., Boucher, É., Bräuning, A., Chatterjee, S., Cherubini, P., Churakova, O.V., Corona, C. and Gennaretti, F., 2021. The influence of decision-making in tree ring-based climate reconstructions. *Nature communications*, *12*(1), pp.1-10.

Charlton‑Perez, A.J., Baldwin, M.P., Birner, T., Black, R.X., Butler, A.H., Calvo, N., Davis, N.A., Gerber, E.P., Gillett, N., Hardiman, S. and Kim, J., 2013. On the lack of stratospheric dynamical variability in low‑top versions of the CMIP5 models. *Journal of Geophysical Research: Atmospheres*, *118*(6), pp.2494-2505.

Coupe, J. and Robock, A., 2021. The Influence of Stratospheric Soot and Sulfate Aerosols on the Northern Hemisphere Wintertime Atmospheric Circulation. *Journal of Geophysical Research: Atmospheres*, p.e2020JD034513.

Driscoll, S., Bozzo, A., Gray, L.J., Robock, A. and Stenchikov, G., 2012. Coupled Model Intercomparison Project 5 (CMIP5) simulations of climate following volcanic eruptions. *Journal of Geophysical Research: Atmospheres*, *117*(D17).

Guillet, S., Corona, C., Stoffel, M., Khodri, M., Lavigne, F., Ortega, P., Eckert, N., Sielenou, P.D., Daux, V., Churakova, O.V. and Davi, N., 2017. Climate response to the Samalas volcanic eruption in 1257 revealed by proxy records. *Nature geoscience, 10*(2), pp.123-128.

Luterbacher, J., Werner, J.P., Smerdon, J.E., Fernández-Donado, L., González-Rouco, F.J., Barriopedro, D., Ljungqvist, F.C., Büntgen, U., Zorita, E., Wagner, S. and Esper, J., 2016. European summer temperatures since Roman times. *Environmental research letters, 11*(2), p.024001.

Neukom, R., Barboza, L.A., Erb, M.P., Shi, F., Emile-Geay, J., Evans, M.N., Franke, J., Kaufman, D.S., Lücke, L., Rehfeld, K. and Schurer, A., 2019. Consistent multi-decadal variability in global temperature reconstructions and simulations over the Common Era. *Nature geoscience, 12*(8), p.643.

Polvani, L.M., Banerjee, A. and Schmidt, A., 2019. Northern Hemisphere continental winter warming following the 1991 Mt. Pinatubo eruption: reconciling models and observations. *Atmospheric Chemistry and Physics, 19*(9), pp.6351-6366.

Robock, A. and Mao, J., 1992. Winter warming from large volcanic eruptions. *Geophysical Research Letters, 19*(24), pp.2405-2408.

Robock, A., 2000. Volcanic eruptions and climate. *Reviews of geophysics, 38*(2), pp.191-219.

Schneider, D.P., Ammann, C.M., Otto‐Bliesner, B.L. and Kaufman, D.S., 2009. Climate response to large, high‐latitude and low‐latitude volcanic eruptions in the Community Climate System Model. *Journal of Geophysical Research: Atmospheres, 114*(D15).

Schneider, L., Smerdon, J.E., Büntgen, U., Wilson, R.J., Myglan, V.S., Kirdyanov, A.V. and Esper, J., 2015. Revising midlatitude summer temperatures back to AD 600 based on a wood density network. *Geophysical Research Letters, 42*(11), pp.4556-4562.

Stenchikov, G., Robock, A., Ramaswamy, V., Schwarzkopf, M.D., Hamilton, K. and Ramachandran, S., 2002. Arctic Oscillation response to the 1991 Mount Pinatubo eruption: Effects of volcanic aerosols and ozone depletion. *Journal of Geophysical Research: Atmospheres, 107*(D24), pp.ACL-28.

Toohey, M., Krüger, K., Bittner, M., Timmreck, C. and Schmidt, H., 2014. The impact of volcanic aerosol on the Northern Hemisphere stratospheric polar vortex: mechanisms and sensitivity to forcing structure. *Atmospheric Chemistry and Physics, 14*(23), pp.13063-13079.

Wilson, R., Anchukaitis, K., Briffa, K.R., Büntgen, U., Cook, E., D'arrigo, R., Davi, N., Esper, J., Frank, D., Gunnarson, B. and Hegerl, G., 2016. Last millennium northern hemisphere summer temperatures from tree rings: Part I: The long term context. *Quaternary Science Reviews*, *134*, pp.1-18.

Zhang, H., Werner, J.P., García-Bustamante, E., González-Rouco, F., Wagner, S., Zorita, E., Fraedrich, K., Jungclaus, J.H., Ljungqvist, F.C., Zhu, X. and Xoplaki, E., 2018. East Asian warm season temperature variations over the past two millennia. *Scientific Reports*, *8*(1), pp.1-11.

---

## Author Response (AR2)

**Dear editor and reviewers, thank you for your second review of the paper. We have compiled a point by point response going through all comments of the three reviews. Our comments are given in blue and bold text.**

**Reviewer 1**

I think the authors have addressed the issues raised in previous versions. I still find the conclusions as a whole not that much far-reaching, but the manuscript is useful as it presents a thorough analysis of an ensemble of simulations. This can be helpful to identify the reasons for the long persistence of cooling found in the different proxy data sets.

**Thank you for your constructive comments. We have taken your comments into account in this round of revisions, as answered in detail below.**

I have a few remarks regarding the formulation of some sentences:

1. Line 110 'To get a significant sample' The word significant is here not clear. A sample cannot be significant in the statistical sense - only a test statistics can be . I think the authors mean to ' increase the power of the test'.

**Thanks, we have corrected this according to your comment.**

2. Line 172 'Towards the end of the simulation, period the ensemble shows a larger spread in the ocean heat content than at the beginning of the simulations.' Delete comma after simulation.

**Thanks, corrected.**

3. Line 377 'The study from Zhong et al. (2011) about the onset of the LIA also concluded the response to be depended..' dependent

**Thanks, corrected.**

4. Line 416 'reveals a less good agreement (see Appendix A, Figure A3)' why not say 'worse agreement'?

**Thanks for your comment. We rephrased the sentence.**

5. Line 445 'Taking into account the entire range of ensemble members is therefore important' The authors mean here, I think, an ensemble of simulations. However, the paragraph is about ensemble of reconstructions (citing Büntgen et al.), and therefore, the sentence is unclear

**Thanks for your comment. We mean both ensembles here. Büntgen et al. (2021) show that using different statistical methods on the same data gives different possible temperature reconstructions, which can be approached in the same manner as an ensemble of realizations of a specific model experiment. Therefore, it is important to take into account the range of the ensemble from both climate model simulations and proxy reconstructions. This has been added to the manuscript for clarification.**

*'Taking into account the entire range of ensemble members from both climate model simulations and proxy reconstructions is therefore important.'*

**References:**

Büntgen, U., Allen, K., Anchukaitis, K. J., Arseneault, D., Boucher, É., Bräuning, A., ... & Esper, J. (2021). The influence of decision-making in tree ring-based climate reconstructions. *Nature communications*, *12*(1), 1-10.

**Reviewer 3**

This is a substantially improved revision and the new organization and clear motivation and intent of the study makes it easier to follow the chain of logic and evidence. I thank the authors for addressing these.

Thank you for your constructive comments. We have taken your comments into account in this round of revisions, as answered in detail below.

A few substantial comments:

1. The authors elected in this revision to focus their NH comparison strictly on the Büntgen et al. 2021 reconstruction. I think a stronger manuscript would still have included comparison to other reconstructions - like those of Stoffel and Guillet -- that also go back to the 6th century. I have no desire to cause additional work for the authors, but considering there are multiple reconstructions for this period, evaluating those (as well as taking the range of the Buntgen reconstruction ensemble reconstruction into account) would help better define how uncertainties in reconstruction temperatures in the first millennium affect the comparison with the ensemble of climate models.

Thanks for your comment. We updated Figure 6 in the manuscript, and the corresponding map in the appendix (Fig. A1, see below) taking Büntgen et al. (2021), Stoffel et al (2015), and Guillet et al (2020) into account, which highlights the uncertainties in reconstruction temperatures in the first millennium compared with the ensemble of the climate model. In the Appendix (Figures A2 and A3), we also show a comparison of the modeled temperature response to other proxy reconstructions from trees Esper (2012, Northern Scandinavia), Büntgen et al (2016, Alps and Altai), Luterbacher et al (2016, Europe) and PAGES2k Neukom et al (2019, NH).

[Figure]

New Figure 6. Model - tree-ring comparison for the Northern Hemisphere (NH) extratropics . The model 2m air temperature anomalies are taken for land only, JJA and 40-75N. Climate model mean and the spread of the model ensemble, tree-ring data for NH1 (Stoffel et al., 2015), N-VOLCv2 NH reconstruction (Guillet et al., 2020), and the mean and the ensemble of Büntgen et al. (2021). Anomalies are wrt 1-1850 CE except for the Guillet et al. data which are wrt 500-1850 CE.

[Figure]

**New Figure A1. Locations of the tree-ring sites used in this study from Stoffel et al. (2015, green dots), Guillet et al. (2020, purple dots), and Büntgen et al. (2021, blue dots). Details can be found in Table A2. Tree ring width data is represented by the circles and maximum latewood density by triangles.**

2. On a related note, I think more emphasis could be placed on looking at the range of responses in both reconstruction and simulation ensembles. The authors address this somewhat tangentially in Line 442 to Line 445 -- that 'real life' is just one iteration of possible climate states, just as the members of the simulation ensemble are possible trajectories the climate system could take. As e.g. Zambri et al. 2019 showed for Laki, initial conditions and internal variability can play an important role in how the forced signal is expressed - thus the existence of both an ensemble of climate model simulations AND an ensemble of reconstructions from Buntgen provides the unique opportunity to consider the range of possible temperature responses resulting from the mixing of internal and forced behavior in the climate system and the uncertainties in the reconstruction process together.

**Thanks for your comment. We agree this is an important point and we have elaborated on this in the discussion. In addition, our follow up paper van Dijk et al. (2022, in review) focuses more on the different individual ensemble members, which we also included in the discussion.**

*'Moreover, the study from Büntgen et al. (2021) shows that it is important to also use an ensemble when it comes to tree-ring reconstructions, as the different statistical methods used to analyze the data give different results. Even though the ensemble mean shows a discrepancy with the model simulation after the 536/540 CE eruptions, some ensemble members fall within the range of the model ensemble spread. Overall, the model ensemble shows less variability in particular to the 536/540 CE response but also to other volcanic eruptions than the tree-ring reconstruction ensemble. The reality can be viewed as one iteration of what could have happened under different initial conditions, ocean states and internal variability. Taking into account the entire range of ensemble members from both climate model simulations and proxy reconstructions is therefore important. In our follow*

*up paper (van Dijk et al., 2022, in review) the individual members are analyzed in more detail for Scandinavia.'*

A few minor points and some suggestions for the authors, particularly with respect to ensuring the introduction is clear:

3. Line 8: 'lasting up to 20 years is simulated' - this is a bit unclear as written - 20 years cooling after each eruption? Can you clarify this?

**Thank you for your comment. By 'up to 20 years' we mean that the response time after all of the eruptions combined is ~20 years. In line 162 in the results section the response time for each eruption is given. We have clarified this in the sentence in line 8.**

**'*After the four large eruptions in 536, 540, 574 and 626 CE, a significant mean surface climate response in the NH lasting up to 20 years is simulated.'***

4. Line 24: It may be worth adding that volcanic eruptions are also our only natural analog for solar radiation management, so understanding (and being able to model them) is critical for this as well.

**Thank you for your comment. We feel one has to be a bit careful here, as the dynamical response in the stratosphere can be of opposite sign, so we do not want to simplify too much. We have added the following sentence to the introduction:**

**'*In order to assess what potential impact they could have on future climate, it is important to understand what the climate response to volcanic forcing was in the past, and which mechanisms are involved. In addition, volcanic eruptions are often studied as a natural analog for solar radiation management (Robock et al., 2013).'***

5. Line 26: what is the difference between a cluster eruption and a double eruption? I have heard people suggest that 'double eruption' is potentially misleading (suggesting multiple eruptions from one source), so best to be clear in this sentence.

**Thank you for your comment. We agree that one has to be careful with the phrase 'double eruption', and so we have changed this throughout the manuscript to 'volcanic double event' to clarify it was not necessarily the same volcanic system.**

6. Line 30: I think this sentence is unclear as written: 'a shift in ice-core records lead ...' -- suggest rewording to: ' ... and updated ice core chronologies reveal two sulphur peaks that correspond to eruptions in 536 and 540 CE (Sigl et al. 2015)'.

**Thank you for your comment. We have taken this up together with the comments from reviewer 4 and changed it to:**

**"Furthermore, historic documents reported a dimming of the sun in 536 CE (Stothers, 1984; Rampino et al., 1988). Revised ice core chronologies reveal two nearby sulfate peaks that correspond to eruptions in 536 CE and 540 CE followed by two large eruptions in 574 CE and 626 CE (Baillie, 2008; Sigl et al., 2015)."**

7. Line 32: suggest replacing 'Thus' with 'Based on these records,'.

**Thanks, we have taken this up in our text.**

8. Line 35: suggest removing the sentence 'This all lines up to a cold period that was initiated by volcanic eruptions' and moving the following sentence (' Indeed, four large volcanic eruptions occurred in 536, 540, 574, and 626 CE (Sigl et al., 2015).') earlier in the paragraph, after the sentence 'Thus, this period was called the Late Antiquity Little Ice Age (LALIA)'.

**Thanks for your suggestion, we have removed the sentence and altered the other sentences accordingly, see our answer to comment No. 6.**

9. Line 104 and elsewhere: There is no 'Year 0' (the Gregorian calendar goes from 1 BCE to 1 CE).

**Thanks for your comment. You are correct and we have changed this throughout the manuscript.**

10. Line 252-259: It would be important here to talk, however, about the ensemble range from Buntgen et al. - for 536 and the following decade in particular, there is large range between the reconstruction ensemble members. Most useful would be to talk about the overlapping (or not) ranges of the ensemble climate model simulations and the ensemble reconstruction.

**Thanks for your suggestion. We agree this is an important point and have added a description of both ensemble spreads to the results section, taking the new Figure 6 (see your comment 2 above) into account. We discuss the ensemble ranges in the discussion section, see also our reply to comment #2.**

11. Line 260: omit 'that catches attention'

**Thanks, we have omitted this part of the sentence.**

12. Line 264 suggest 'which may be due to ...'

**Thanks, this has been changed according to your comment.**

13. Line 296: suggest replacing 'A few of the debated points are about discussing' with 'These include'

**Thanks, we have followed your suggestion.**

14. Line 300: 'which could dampen the signal' - can you say more about this? Wouldn't this suggest the signal is unforced then (part of internal variability)?

**Thanks for your question. As we have a mix of tropical and extratropical eruptions including a double event, we probably can not expect a significant signal in the volcanic mean NAO response (Schneider et al., 2009). We have clarified this and added the following instead:**

*'which may explain the lack of a significant NAO response'.*

15. Line 328: suggest replacing 'large noise' with 'large internal variability'

**Thanks, we have taken up your comment and changed the sentence accordingly.**

16. Lines 409-410: I'm not sure how to reconcile the observation that the match is good for 'the recovery time of ~20 year after the peak cooling' but also that the match is bad for 'the lag after the 536/540 CE eruptions'? These seem like two contradictory statements?

**Thank you for the comment, we see your point that this is a contradiction. We have changed the sentence according to the new Figure 6 above.**

*'The model - tree-ring comparison with the Büntgen et al. (2021) tree-ring ensemble reconstruction (Fig. 6) shows a very good agreement in the timing of the peak cooling of the 2 m air temperature anomalies for the NH extratropics land only JJA. The mismatches that are still present in this NH comparison, like the strength of the peak cooling, as well as the lag after the 536/540 CE eruptions, include potential deficiencies and uncertainties regarding the method. For example, for the Büntgen et al. (2021) tree ring reconstructions TRW was used, which is known to give a lagged and smoothed response (e.g. Esper et al., 2015; Zhang et al., 2015; Lücke et al., 2019; Zhu et al., 2020). This could explain the lag after the volcanic double event for Büntgen et al. (2021), whereas the other two reconstructions, Stoffel et al. (2015) and Guillet et al. (2020), are more in line with the model simulations.'*

17. Line 412: ' Reconstructions are becoming more uncertain ...' - this is true in general, although not exactly relevant to the current paper, where the number of chronologies in Buntgen is the same going back through the 6th century.

**Indeed. Here we refer to reconstructions in general, which are becoming more uncertain back in time, to explain why we chose not to include certain other reconstructions in the main manuscript. We have clarified this point by adding the following:**

*'... This could explain the lag after the double eruption event for Büntgen et al. (2021), whereas the other two reconstructions, Stoffel et al. (2015) and Guillet et al. (2020), are more in line with the model simulations. As reconstructions are becoming more uncertain further back in time due to the sparseness of available proxy records, which mainly rely on tree ring records. This is the case especially before 1200 CE (Masson-Delmotte, 2013; Esper et al., 2018; Neukom et al., 2019) and therefore, we have chosen to use the Büntgen et al. (2021) reconstruction as it uses the same number of tree ring records throughout the entire CE. Additionally, we have chosen to include the reconstructions by Stoffel et al. (2015) and Guillet et al. (2020) as they both consist of a mix of TRW and MXD records. Testing a comparison of the model results with multiproxy temperature reconstructions for the entire NH annual mean (Neukom et al., 2019) reveals a worse agreement (see Appendix A, Figure A3).'*

18. Line 428: 'do not agree as for' - I'm unclear what this means? Does this mean the specific comparison is not as good as the NH comparison, or that they are similarly bad? Please clarify.

**Thanks, here we mean that the specific comparison is not as good as for the NH comparison. We have clarified this in the text.**

'*Comparing these specific sites, the model and tree-ring reconstruction do not agree as well as they do for the NH tree ring ensemble reconstructions …*'

19. Line 449: As in the my previous review, I don't think this is at all relevant - if anything the cooling would have made the trees better recorders of temperature and there is no reason in this case to suspect a role for moisture anomalies specifically in the local mismatch between models and trees.

**After our first round of revision, we added the papers of Basset et al. (1964) and Müller et al. (2016) to support the theory that moisture availability could have impacted the tree-ring growth. The Alps do indeed show a slight increase in simulated precipitation for the 20 year mean, but they also have an increase in evaporation at the same time. We find the atmospheric circulation separation at 40/45°N latitude very interesting, and thus would like to leave it in the paper to stimulate further work and discussion within the paleo climate proxy record community.**

**References:**

**Bassett, J. R. (1964). Tree growth as affected by soil moisture availability. *Soil Science Society of America Journal*, *28*(3), 436-438.**

**Büntgen, U., Myglan, V. S., Ljungqvist, F. C., McCormick, M., Di Cosmo, N., Sigl, M., ... & Kirdyanov, A. V. (2016). Cooling and societal change during the Late Antique Little Ice Age from 536 to around 660 AD. *Nature geoscience*, *9*(3), 231-236.**

**Büntgen, U., Allen, K., Anchukaitis, K. J., Arseneault, D., Boucher, É., Bräuning, A., ... & Esper, J. (2021). The influence of decision-making in tree ring-based climate reconstructions. *Nature communications*, *12*(1), 1-10.**

**Esper, J., Büntgen, U., Timonen, M., & Frank, D. C. (2012). Variability and extremes of northern Scandinavian summer temperatures over the past two millennia. *Global and Planetary Change*, *88*, 1-9.**

**Esper, J., Schneider, L., Smerdon, J. E., Schöne, B. R., & Büntgen, U. (2015). Signals and memory in tree-ring width and density data. *Dendrochronologia*, *35*, 62-70.**

**Guillet, S., Corona, C., Ludlow, F., Oppenheimer, C., & Stoffel, M. (2020). Climatic and societal impacts of a "forgotten" cluster of volcanic eruptions in 1108-1110 CE. *Scientific reports*, *10*(1), 1-10.**

**Lücke, L. J., Hegerl, G. C., Schurer, A. P., & Wilson, R. (2019). Effects of memory biases on variability of temperature reconstructions. *Journal of Climate*, *32*(24), 8713-8731.**

**Luterbacher, J., Werner, J. P., Smerdon, J. E., Fernández-Donado, L., González-Rouco, F. J., Barriopedro, D., ... & Zerefos, C. (2016). European summer temperatures since Roman times. *Environmental research letters*, *11*(2), 024001.**

**Müller, M., Schwab, N., Schickhoff, U., Böhner, J., & Scholten, T. (2016). Soil temperature and soil moisture patterns in a Himalayan alpine treeline ecotone. *Arctic, Antarctic, and Alpine Research*, *48*(3), 501-521.**

Neukom, R., Barboza, L. A., Erb, M. P., Shi, F., Emile-Geay, J., Evans, M. N., ... & von Gunten, L. (2019). Consistent multi-decadal variability in global temperature reconstructions and simulations over the Common Era. *Nature geoscience*, *12*(8), 643.

Robock, A., MacMartin, D. G., Duren, R., & Christensen, M. W. (2013). Studying geoengineering with natural and anthropogenic analogs. *Climatic Change*, *121*(3), 445-458.

Schneider, D. P., Ammann, C. M., Otto-Bliesner, B. L., & Kaufman, D. S. (2009). Climate response to large, high-latitude and low-latitude volcanic eruptions in the Community Climate System Model. *Journal of Geophysical Research: Atmospheres*, *114*(D15).

Stoffel, M., Khodri, M., Corona, C., Guillet, S., Poulain, V., Bekki, S., ... & Masson-Delmotte, V. (2015). Estimates of volcanic-induced cooling in the Northern Hemisphere over the past 1,500 years. *Nature Geoscience*, *8*(10), 784-788.

van Dijk, E., Mørkestøl Gundersen, I., de Bode, A., Høeg, H., Loftsgarden, K., Iversen, F., ... & Krüger, K. (2022). Climate and society impacts in Scandinavia following the 536/540 CE volcanic double event. *Climate of the Past Discussions*, 1-55.

Zhang, H., Yuan, N., Esper, J., Werner, J. P., Xoplaki, E., Büntgen, U., ... & Luterbacher, J. (2015). Modified climate with long term memory in tree ring proxies. *Environmental Research Letters*, *10*(8), 084020.

Zhu, F., Emile-Geay, J., Hakim, G. J., King, J., & Anchukaitis, K. J. (2020). Resolving the differences in the simulated and reconstructed temperature response to volcanism. *Geophysical Research Letters*, *47*(8), e2019GL086908.

**Reviewer 4**

Dear authors,
Thank you for the revisions which have improved the paper. For the submission to become acceptable, more work is needed as some of the new text elements do not fit nicely into the previous version which makes it difficult to follow all ideas. I would also strongly encourage the authors to have their paper edited by a native speaker.
The most relevant comments are listed below:

*Thank you for your thorough comments. We have arranged the flow of the text elements and the manuscript was checked by a native speaker for English grammar.*

*We had some problems with your line numbering listed below, as your comments do not seem to match the lines that are referred to in those comments. We have tried our best to interpret to which section the comments apply, and responded accordingly below.*

1. lines 41-42: One major double event leading to widespread and severe cooling occurred in 1108-10. Please add the following citation: Guillet, S., Corona, C., Ludlow, F.M., Oppenheimer, C., Stoffel, M., 2020. Climatic and societal impacts of a "forgotten" cluster of volcanic eruptions in 1108-1110 CE. Nature Scientific Reports 10, 6715.

*Thanks for your suggestion. We have added this reference to the introduction.*

2. line 46: Sigl et al. (2015) rely on tree-ring data that were - at least partly - published in Büntgen et al. (2011). I suggest to remove the Sigl reference here as that paper used ice cores primarily.

*Thank you for your suggestion. We have removed the Sigl et al. (2015) in this context.*

'*One of the coldest decades of the last 2000 years in the NH and Europe is visible in tree-ring records during the mid-sixth century (Larsen et al., 2008; Büntgen et al., 2011).*'

3. line 47ff: This is a bit messy in my view. The ice-core data need to be put in context more carefully. Also, the reference to four eruptions (lines 54-55) come somewhat late.

*Thanks for your comment. We have restructured the paragraph including Reviewers 1 and 3 comments as well as follows:*

*…"Several cluster eruptions and volcanic double events occurred in the last 2000 years as recorded in the ice core records, coinciding with cold periods in Northern Hemisphere (NH) tree-ring records (Briffa et al., 1998; Sigl et al., 2013). One of the coldest decades of the last 2000 years in the NH and Europe is visible in tree-rings during the mid-sixth century, which was preceded by two volcanic eruptions as recorded in ice cores (Larsen et al., 2008; Büntgen et al., 2011). Furthermore, historic documents reported a dimming of the sun in 536 CE (Stothers, 1984; Rampino et al., 1988). Revised ice core chronologies reveal two sulfate peaks that correspond to eruptions in 536 CE and 540 CE followed by two large eruptions in 574 CE and 626 CE (Baillie, 2008; Sigl et al., 2015). Reconstructed tree-ring temperatures from the Alps and Altai show a century-long cooling that might have exceeded that of the Little Ice Age (LIA) during the 14th-19th century (Büntgen et al., 2016). Based on these*

**records, this period was called the Late Antiquity Little Ice Age (LALIA). wever, other studies reveal contrasting results35**
**on how long lasting the surface cooling in the NH extratropics was, varying from multi-decadal to centennial cooling. These results are based on tree ring reconstruction methods and tree ring record updates, as well as ice-core records and documentary evidence (e.g. Esper et al 2012b; Matskovsky and Helama, 2014; Helama et al., 2017; Guillet et al., 2020; Büntgen et al., 2021; Helama et al., 2021). Thus, the duration of this volcanic induced cooling event remains open."...**

4. lines 170-1: Setting all eruptions to January will have an influence on the cooling and the persistence of cold conditions. This has to be said more clearly and - ideally - remediated. Using different seasonalities would certainly help to determine the spread of cooling and uncertainties.

**We are using the volcanic forcing data set eVolv2k (Toohey and Sigl, 2017), which has been recommended for CMIP6/PMIP4 past2k simulations (Jungclaus et al., 2017)). This volcanic forcing sets January as the standard eruption month for unknown eruptions. Upcoming CMIP7/PMIP5 volcanic forcing datasets will remediate this, where our study is contributing to. Indeed, the eruption season is important for understanding the climate response, as clearly mentioned in the discussion in line 405. For the 536 CE eruption a winter eruption is consistent with the timing of the dust veil observed over Europe in March (Stothers, 1984). Sensitivity experiments with different seasons have been carried out elsewhere (Toohey et al., 2011). We have clarified this in the discussion.**

*'In addition, the eruption season is also important, as different eruption seasons give different atmospheric circulation patterns and therefore influence the transport of the sulfate aerosol, leading to different surface responses (Toohey et al., 2011).'*

5. lines 179ff: Relying exclusively on tree-ring width data (as presented in Büntgen et al. 2021) is problematic. Several of the NH reconstructions extend back to 500 CE and should be kept as they contain latewood density data which are less affected by memory in tree rings.

**Yes we agree. Thus, we have added the NH reconstructions by Stoffel et al. (2015) and Guilett et al. (2020), which also include MXD data, to Figure 6, as also suggested by Reviewer 3. In addition, we have added a discussion to the reconstructions from Matskovsky and Helama, 2014 and Helama et al., 2021 as suggested by you below as well. The text has been adapted accordingly in the manuscript including MXD data in the introduction, method, results and the discussion sections.**

**Tree ring data (Methods):**
*'For the model-tree ring temperature comparison, different tree ring data and reconstructions are used. The tree ring data used are based on tree ring width (TRW) and maximum latewood density (MXD). TRW is known to have biological memory and gives a lagged and smoothed response to volcanic eruptions. In contrast, MXD is based on cell density, which gives a better representation of volcanic induced surface cooling (Anchukaitis et al., 2012; Esper et al., 2015; Zhu et al., 2020; Ludescher et al., 2020). MXD data is therefore the preferred target for our model comparison if available, but the data is sparse during the first millennium. Thus, both tree ring methods are taken into account using the reconstructions from Stoffel et al. (2015), Guilett et al. (2020), and Büntgen et al. (2021), next to others (see Appendix A). Stoffel et al. (2015) and Guilett et al. (2020) consist of a mix of MXD and TRW. The tree ring ensemble reconstruction from Büntgen et al. (2021) is based on TRW only and is taken from 9 sites over the NH covering the past 2000 years (Fig. A1). The raw data were distributed to 15 different dendrochronology groups. These*

*groups all carried out their own statistical methods on the data, after which the now different data sets were combined to form a tree ring ensemble. More details about the data and the reconstruction method can be found in the corresponding publications.*
*To use the same reference period for the model and tree ring data, we subtracted the 1-1850 CE mean from the model and tree ring ensemble. For the model-tree ring comparison a land mask was applied to the model 2 m air temperature and we analyzed only the NH extratropics between 40∘and 75N. The tree ring data sets capture the boreal summer temperature during June, July and August (JJA) and were therefore compared to JJA 2 m air temperatures from the model.'*

6. lines 188ff: Yes, you are right that MXD data is sparse but leaving it out completely would be the wrong decision. I also suggest that the authors check the following publication and discuss results therein with their findings: Helama, S., Stoffel, M., Hall, R.J., Jones, P.D., Arppe, L., Matskovsky, V.V., Timonen, M., Nöjd, P., Mielikäinen, K., Oinonen, M., 2021. Recurrent transitions to Little Ice Age-like climatic regimes over the Holocene. Climate Dynamics 6, 3817–3833.

**Thanks for your suggestion and see our responses to include MXD data above. We have added a discussion of our results with regard to the Helama et al. (2021) and Matskovsky and Helama (2014) papers in the tree ring section (new line 454). Next, we have added Helama et al. (2021) to the discussion on atmospheric circulation patterns (new line 350).**

*'In addition, Matskovsky and Helama (2014) and Helama et al. (2021) report a century long cooling for Northern Scandinavia from 530 to 650 CE based on MXD, TRW and δ13C data, which is not supported by our model simulations (Fig. A2a). Other proxy-based studies (Larsen et al., 2008; Esper et al., 2012b; Luterbacher et al., 2016; Helama et al., 2017; Neukom et al., 2019) found a cooling up to 570 CE for Europe, Scandinavia, and the NH, based on tree ring, ice-core, lake sediment, and documentary records (Figs. A2a and d, and A3). Comparisons for the different tree ring data sets have been carried out by previous studies for the last millennium (Wilson et al., 2016; Lücke et al., 2019). However, not enough records exist for the first millennium to carry out a similar comparison for this period yet. From the perspective of our model results, the persistence of the cooling was not as long lasting as the tree ring sites from the Alps, Altai, and Northern Scandinavia suggest. … Another possibility is, that our model resolution is too coarse to fully capture the topography of the Alps, Altai, and Northern Scandinavia…'*

*'Helama et al. (2021) describe an East Atlantic pattern during the study period, corresponding to clear sky conditions over Northern Scandinavia, as obtained from tree ring proxies (TRW, MXD, and stable carbon isotope (δ13C)). This pattern of reduced cloudiness is consistent with the dry conditions simulated over Northern Europe in our model simulations.'*

7. line 325: what is the cooling obtained in NH reconstructions based on MXD or mixed MXD-TRW reconstructions? It would be nice to have these as well.

[Figure]

**New Figure 6. Model - tree-ring comparison for the Northern Hemisphere (NH) extratropics. The model 2m air temperature anomalies are taken for land only, JJA and 40-75N. Climate model mean and the spread of the model ensemble, tree-ring data for NH1 (Stoffel et al., 2015), N-VOLCv2 NH reconstruction (Guillet et al., 2020), and the mean and the ensemble of Büntgen et al. (2021). Anomalies are wrt 1-1850 CE except for the Guillet et al. data which are wrt 500-1850 CE.**

**Yes, we agree and we have updated Figure 6 in the manuscript (see new figure 6) and the corresponding text. See our previous comments to you and Reviewer 3. MXD records do not exist for the entire NH, only mixed TRW and MXD reconstructions are available. However, Wilson et al. (2016) as well as Lücke et al. (2019) provide a comparison between modeled, TRW, MXD and mixed reconstructions for the last millennium. For a comparison like this to be possible for the first millennium requires more investment in tree-ring records for the NH during this period. We have added these two references now to the discussion.**

'*Comparisons for the different tree ring data sets have been carried out by previous studies for the last millennium (Wilson et al., 2016; Lücke et al., 2019). However, not enough records exist for the first millennium to carry out a similar comparison for this period yet.*'

8. line 326ff: TRW data have biological memory, so comparing persistence is of limited use. You would better compare with the available NH MXD records here and put the TRW, MXD, Neukom and model results into perspective.

**Thank you for your comment. We have included this comparison now and updated Figure 6 in the manuscript and the corresponding text. See also our previous comments to you and Reviewer 3.**

9. line 334: Why do you assume January for the eruption if documentary evidence suggests an autumn eruption?

**We assume the reviewer is referring to the 626 CE eruption and the reason for the mismatch between the model simulations and the tree-ring records. See also our answer to your comment #4 above. For the volcanic forcing the eVolv2k data set (Toohey and Sigl, 2017) was used following the CMIP6/PMIP4 protocol (Jungclaus et al., 2017), January is used as the standard eruption month when the eruption is unknown. Upcoming CMIP7/PMIP5 volcanic forcing datasets will remediate this, where our study is contributing to.**

**References:**

Guillet, S., Corona, C., Ludlow, F., Oppenheimer, C., & Stoffel, M. (2020). Climatic and societal impacts of a "forgotten" cluster of volcanic eruptions in 1108-1110 CE. *Scientific reports*, *10*(1), 1-10.

Helama, S., Stoffel, M., Hall, R. J., Jones, P. D., Arppe, L., Matskovsky, V. V., ... & Oinonen, M. (2021). Recurrent transitions to Little Ice Age-like climatic regimes over the Holocene. *Climate dynamics*, *56*(11), 3817-3833.

Jungclaus, J. H., Bard, E., Baroni, M., Braconnot, P., Cao, J., Chini, L. P., ... & Zorita, E. (2017). The PMIP4 contribution to CMIP6–Part 3: The last millennium, scientific objective, and experimental design for the PMIP4 past1000 simulations. *Geoscientific Model Development*, *10*(11), 4005-4033.

Lücke, L. J., Hegerl, G. C., Schurer, A. P., & Wilson, R. (2019). Effects of memory biases on variability of temperature reconstructions. *Journal of Climate*, *32*(24), 8713-8731.

Matskovsky, V. V., & Helama, S. (2014). Testing long-term summer temperature reconstruction based on maximum density chronologies obtained by reanalysis of tree-ring data sets from northernmost Sweden and Finland. *Climate of the Past*, *10*(4), 1473-1487.

Sigl, M., Winstrup, M., McConnell, J. R., Welten, K. C., Plunkett, G., Ludlow, F., ... & Woodruff, T. E. (2015). Timing and climate forcing of volcanic eruptions for the past 2,500 years. *Nature*, *523*(7562), 543-549.

Stoffel, M., Khodri, M., Corona, C., Guillet, S., Poulain, V., Bekki, S., ... & Masson-Delmotte, V. (2015). Estimates of volcanic-induced cooling in the Northern Hemisphere over the past 1,500 years. *Nature Geoscience*, *8*(10), 784-788.

Stothers, R. B. (1984). Mystery cloud of AD 536. *Nature*, *307*(5949), 344-345.

Toohey, M., Krüger, K., Niemeier, U., & Timmreck, C. (2011). The influence of eruption season on the global aerosol evolution and radiative impact of tropical volcanic eruptions. *Atmospheric Chemistry and Physics*, *11*(23), 12351-12367.

Toohey, M., & Sigl, M. (2017). Volcanic stratospheric sulfur injections and aerosol optical depth from 500 BCE to 1900 CE. *Earth System Science Data*, *9*(2), 809-831.

Wilson, R., Anchukaitis, K., Briffa, K. R., Büntgen, U., Cook, E., D'arrigo, R., ... & Zorita, E. (2016). Last millennium northern hemisphere summer temperatures from tree rings: Part I: The long term context. *Quaternary Science Reviews*, *134*, 1-18.

---

## Author Response (AR3)

Dear authors, thank you very much for the substantial revisions and taken the comments of the reviewers well into account. I only have three minor issues I wish to address, this can be done in a very short time.

Concerning text lines 290ff (document with track changes):
I refer to my comment from December where I stated on the Buentgen/model comparison:
It seems that there is a large difference around 540, around 575 and 630, also obvious is the general overestimation from around 545 to 575 and underestimation until 640. Indeed, it would be interesting if this is a feature for this reconstruction only or also shared with independent evidence from other reconstructions.

you very well addressed the discrepancies around 540, not on the other two periods. Could you please check again your text and see if that can be addressed.

Dear Jürg Luterbacher, thank you for your valuable comments.

1.) We have changed the paragraph with the discrepancies for the other periods in the results section (line 274-278 in the new manuscript):

'The temperature recovery is different for the Büntgen et al. (2021) tree ring ensemble and the model ensemble (Fig. 6). After the 536/540 CE volcanic double event, the modeled temperature recovers faster than the reconstructed Büntgen et al. (2021) mean temperature, in agreement with Stoffel et al. (2015) and Guillet et al. (2020). However, even though the Büntgen et al. (2021) ensemble mean falls outside the model ensemble range after 536/540 CE, some of the members from the reconstructions are still within the model ensemble spread. The model and tree ring data comparison shows a quite diverse picture dependent on the time period and reconstruction considered. In the three decades after the 536/540 CE volcanic double event, from around 545 CE to 575 CE the simulated temperature anomalies are in good agreement with Stoffel et al (2015) and Guillet et al (2020), but smaller in comparison to Büntgen et al (2021). Around 640 CE the model results and the reconstructed Büntgen et al (2021) data agree quite well, while the Stoffel et al (2015) and Guillet et al (2020) data show a distinct minimum."

2.) Please refer to Figure 6 in the text when you start describing from line 290ff

Thanks, we have added the reference to Figure 6 in the text.

3.) Generally on the period under considerations, please could you also cite the following new peer reviewed contribution related to the topic:
Xoplaki, E., et al., 2021: Hydrological Changes in Late Antiquity: Spatio-Temporal Characteristics and Socio-Economic Impacts in the Eastern Mediterranean. In: Erdkamp P., Manning J.G., Verboven K. (eds) Climate Change and Ancient Societies in Europe and the Near East. Palgrave Studies in Ancient Economies. Palgrave Macmillan, Cham. https://doi.org/10.1007/978-3-030-81103-7_18, pp 533-560

We have added this reference to the discussion about the Mediterranean (line 354 in the new manuscript).

'Xoplaki et al. (2021) reconstructed the hydroclimate from speleothems and lake sediments for the eastern Mediterranean, and found a sharp change to wetter conditions in the second half of the 6th century for all sites.'